# On the Effect of Defection in Federated Learning and How to Prevent It

## Abstract

Federated learning is a machine learning protocol that enables a large population of agents to collaborate. These agents communicate over multiple rounds to produce a single, consensus model. Despite this collaborative framework, there are instances where agents may choose to defect permanently—essentially withdrawing from the collaboration—if they are content with their instantaneous model in that round. This work demonstrates the detrimental impact such defections can have on the final model's robustness and ability to generalize. We also show that current federated optimization algorithms fall short in disincentivizing these harmful defections. To address this, we introduce a novel optimization algorithm with theoretical guarantees to prevent defections while ensuring asymptotic convergence to an effective solution for all participating agents. We also provide numerical experiments to corroborate our findings and demonstrate the effectiveness of our algorithm.

## 1 Introduction

Collaborative machine learning protocols have not only fueled significant scientific discoveries (Bergen & Petryshen, 2012) but are also gaining traction in diverse sectors like healthcare networks (Li et al., 2019; Powell, 2019; Roth et al., 2020), mobile technology (McMahan & Ramage, 2017; Apple; Paulik et al., 2021), and financial institutions (Shiffman et al., 2021). A key factor propelling this widespread adoption is the emerging field of federated learning (McMahan et al., 2016b). Federated Learning allows multiple agents (also called devices) and a central server to tackle a learning problem collaboratively without exchanging or transferring any agent's raw data, generally over a series of communication rounds.

Federated learning comes in various forms, ranging from models trained on millions of peripheral devices like Android smartphones to those trained on a limited number of large data repositories. The survey by Kairouz et al. (2019) categorizes these two contrasting scales of collaboration, which come with distinct system constraints, as *"cross-device"* and *"cross-silo"* federated learning, respectively. This paper concentrates on a scenario that falls between these two extremes. For example, consider a nationwide medical study led by a government agency to explore the long-term side effects of COVID-19. This agency selects several dozen hospitals to partake in the study to develop a robust model applicable to the entire national populace. Specifically, the server (the governmental agency) has a distribution $\mathcal{P}$ over agents (hospitals), and each agent $m$ maintains a local data distribution $\mathcal{D}_m$ (the distribution of its local patients). The server's goal is to find a model $w$ with low **population loss**, i.e.,

$$\mathbb{E}_{m\sim\mathcal{P}}[F_m(w) := \mathbb{E}_{z\sim\mathcal{D}_m}[f(w;z)]], \tag{1}$$

where $f(w;z)$ represents the loss of model $w$ at data point $z$.

Due to constraints like communication overhead, latency, and limited bandwidth, training a model across all agents (say all the hospitals in a country) is infeasible. The server, therefore, aims to achieve its goal by sampling $M$ agents from $\mathcal{P}$ and minimizing the "average loss", given by

$$F(w) := \frac{1}{M}\sum_{m=1}^{M} F_m(w). \tag{2}$$

If the sample size $M$ is sufficiently large and the server can identify a model with low average loss, likely, the model will also have low population loss, provided the data is only moderately

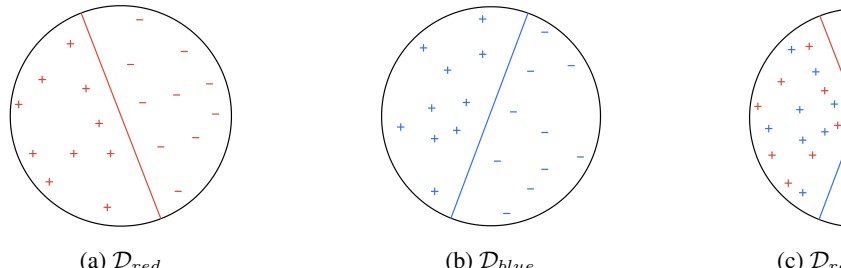

(a) $\mathcal{D}_{red}$        (b) $\mathcal{D}_{blue}$        (c) $\mathcal{D}_{red} \cup \mathcal{D}_{blue}$

Figure 1: Consider two agents {red, blue} with distributions $\mathcal{D}_{red}$ and $\mathcal{D}_{blue}$ on the ball in $\mathbb{R}^2$. Figures 1a and 1b depict these distributions, where the number of $+$ and $-$ represent point masses in the density function of each distribution. Note that each distribution has multiple zero-error linear classifiers, but we depict the max-margin classifiers in plots (a) and (b). Figure 1c shows the combined data from these agents and the best separator (black), classifying the combined data perfectly. However, the red (blue) separator on the blue (red) data has a higher error rate of 20 percent. Thus, if either agent defects during training, any algorithm converging to the max-margin separator for the remaining agent will incur an average 10 percent error.

heterogeneous. Hence, our objective (from the agency's perspective) is to find a model with a low average loss. When all $M$ agents share some optima—meaning a model exists that satisfies all participating hospitals—such a model is also universally desirable. However, this paper reveals that this collaborative approach falters when agents aim to reduce their workload or, in the context of hospitals, aim to minimize data collection and local computation.

More specifically, prevalent federated optimization techniques like federated averaging (FEDAVG) (McMahan et al., 2016a) and mini-batch SGD (Dekel et al., 2012; Woodworth et al., 2020) (refer to appendix A), employ a strategy of intermittent communication. Specifically, for each round $r = 1, \ldots, R$:

- **Server-to-Agent Communication:** The server disseminates a synchronized model $w_{r-1}$ to all participating agents;
- **Local Computation:** Every device initiates local training of its model from $w_{r-1}$ using its own dataset;
- **Agent-to-Server Communication:** Each device transmits the locally computed updates back to the server;
- **Model Update:** The server compiles these updates and revises the synchronized model to $w_r$.

If all agents provide their assigned model updates and collaborate effectively, the synchronized model should converge to a final model $w_R$, with a low average loss $F(w_R)$. However, agents face various costs, such as data collection, computational effort, and potential privacy risks. Thus, rational agents might exit the process once they are content with their current model's performance on their local data. For instance, in our example, hospitals may prioritize a model that performs well solely for their local patient demographics.

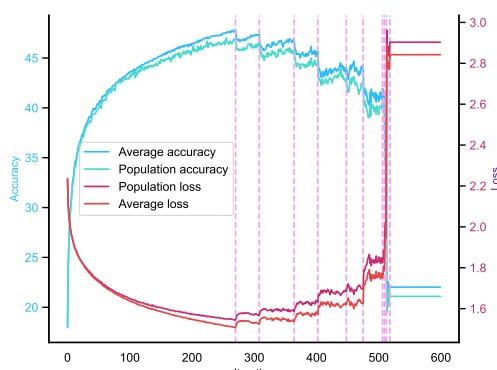

Figure 2: Impact of defections on both average and population accuracy metrics when using federated averaging with local update steps $K = 5$ and step size $\eta = 0.1$. The CIFAR10 dataset (Krizhevsky et al., 2009) is processed to achieve a heterogeneity level of $q = 0.9$ (refer to appendix D for data generation details). Agents aim for a precision/loss threshold of $\epsilon = 0.2$, and dashed lines indicate the moments when an agent defects. It is evident that each defection adversely affects the model's accuracy. For example, the peak average accuracy drops from approximately 46% prior to any defections to around 22% after 500 iterations. A similar decline is observed in population accuracy.

***Defections***, or the act of permanently exiting before completing $R$ rounds, can adversely affect the quality of the final model $w_R$. This occurs because defecting agents make their data unavailable, leading to a loss of crucial information for the learning process.

The impact is particularly significant if the defecting agent had a large or diverse dataset or was the sole contributor to a specific type of data (refer to Figure 1). Consequently, the final model $w_R$ may fail to achieve a low average loss $F(w_R)$. For empirical evidence, see Figures 2 and 3a, which show the effect of defections on the average objective. Defections can generally lead to the following issues:

- **Suboptimal Generalization.** Even if not all data is lost due to defections, the resulting dataset may become imbalanced. This can happen if the remaining agents do not adequately represent the broader population or their updates are too similar. The model may overfit the remaining data, leading to poor performance for unsampled agents from $\mathcal{P}$. This is experimentally simulated in Figure 3b.

- **Inconsistent Final Performance.** When agents belong to protected groups (e.g., based on gender or race), it is crucial for the model to be fair across these groups (Ezzeldin et al., 2021). However, defections can result in a model that performs poorly for some agents. The effects of defection on the performance of the best, worst, and median agents are illustrated in Figure 3c.

- **Disproportionate Workload.** Even if defections do not directly harm the model's performance, they can increase the workload for the remaining agents. These agents may need to provide additional updates to compensate for the lost data, leading to increased

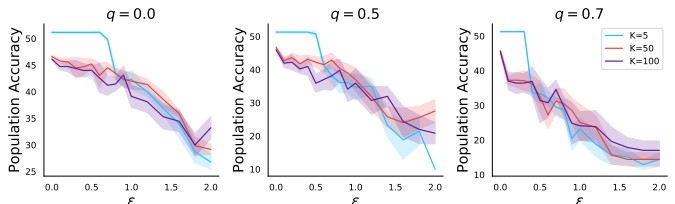

(a) Effect of defection on average accuracy.

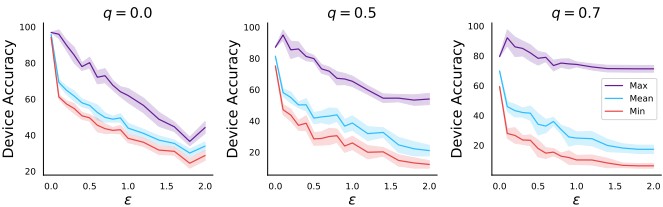

(b) Effect of defection on population accuracy.

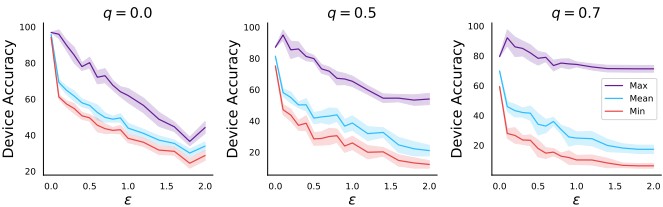

(c) Effect of defection on the min, mean, and max device accuracies.

Figure 3: Our study examines how variations in data heterogeneity $q$ (across different plots at each row), required precision $\epsilon$ (on the $x$-axis in each plot), and the number of local update steps $K$ (different curves in plots 3a and 3b, while $K$ is fixed as 100 in 3c) affect multiple accuracy metrics impacted by agents' defection behavior during training with federated averaging (Algorithm 3). In Figure 3a, we plot the final model's accuracy for the average objective. In Figure 3b, we plot the accuracy for the population objective $\mathbb{E}_{m\sim\mathcal{P},z\sim\mathcal{D}_m}[f(.;z)]$. This captures the generalization ability of the final model. In Figure 3c, we plot the min, max, and mean device accuracy out of the $M$ devices participating in training for the final model. As $\epsilon$ increases, the likelihood of each device defecting increases, so all the curves almost always decrease. We also note that on increasing the heterogeneity of the task, the accuracies all decrease while the variance between different devices increases. The task is multi-class classification on CIFAR-10. We plot the accuracy instead of the loss value for better interpretability. More details are in appendix D. All experiments are repeated ten times, and the plotted curves are averaged across the runs with error bars for $95\%$ confidence level.

latency and bandwidth usage. This is particularly problematic if agents are geographically dispersed and can discourage them from continued participation due to resource constraints.

Besides the aforementioned issues, defections can also lead to unpredictable outcomes when agents generate data in real-time (Huang et al., 2021; Patel et al., 2023), or when they undergo distribution shifts, as seen in self-driving cars (Nguyen et al., 2022). Inspired by these observations, our work aims to address the following key questions:

*1. Under what conditions are defections detrimental to widely-used FL algorithms like FEDAVG?*

*2. Is it possible to develop an algorithm that mitigates defections while still optimizing effectively?*

**Contributions.** In Section 2, we provide a formal framework for understanding defections in federated learning, which often arise due to agents' desire to minimize computational and communication overhead. Specifically, agents will likely exit the training process once they obtain a satisfactory model. In Section 3, we distinguish between benign and harmful defections and explore the influence of (i) **initial conditions**, (ii) **learning rates**, and (iii) **aggregation methods** on the occurrence of harmful defections. Our findings indicate that simply averaging local updates is insufficient to prevent harmful defections. We further validate our theoretical insights with empirical studies (see Figures 2 and 3), confirming that defections can substantially degrade the performance of the final model.

Lastly, in Section 4, we introduce our novel algorithm, ADA-GD. Under mild (and possibly necessary) conditions, this algorithm converges to the optimal model for problem (2) without any agent defecting. Unlike simple averaging methods used in FEDAVG and mini-batch SGD, our approach tailors the treatment for each device. For devices on the verge of defecting, we utilize their gradient information to define a subspace where the gradients of the remaining devices are projected. This newly aggregated gradient is then employed to update the current model. This nuanced update strategy is designed to improve the average objective while preventing defections, making the algorithm complex to analyze. We tackle this complexity through a unique case-based analysis. We also empirically contrast our algorithm with FEDAVG, showing that our method effectively prevents defections and results in a superior final model (see Figure 7).

## 1.1 RELATED WORK

The complexity of managing defections in federated learning arises primarily from two factors: (i) the ongoing interactions between the server and the agents, allowing devices to access all intermediate models, and (ii) the fact that agents do not disclose their raw data to the server. This results in an information asymmetry, as the server can only speculate about potential defections and has no way to retract an intermediate model already exposed to an agent. This sets our problem apart as particularly challenging, and no current theoretical research aims to mitigate agent defections while solving problem (2). Most existing studies concentrate on single-round interactions (Karimireddy et al., 2022; Blum et al., 2021), which essentially incentivize agents to gather and relinquish samples to the server. The MW-FED algorithm by Blum et al. (2017) aligns with the intermittent communication model and deliberately decelerates the advancement of agents nearing their target accuracy levels. This makes it potentially useful for preventing defections (Blum et al., 2021). However, its applicability in our context remains unclear, and no convergence analysis exists for this method in heterogeneous settings. We provide an exhaustive review of these and other related works, including those concerning fairness in federated learning, in appendix B.

## 2 PROBLEM SETUP

Recall that our (server's/government agency's) learning goal is to minimize $F(w) = \frac{1}{M} \sum_{m \in [M]} F_m(w)$ over all $w \in \mathcal{W} \subseteq \mathbb{R}^d$, where $F_m(w) := \mathbb{E}_{z \sim \mathcal{D}_m}[f(w; z)]$ is the loss on agent $m$'s data distribution. We assume the functions $F_m$'s are differentiable, convex, Lipschitz, and smooth.

**Assumption 1.** *Differentiable $F_m : \mathcal{W} \to \mathbb{R}$ is convex and $H$-smooth, if for all $u, v \in \mathcal{W}$,*

$$F_m(v) + \langle \nabla F_m(v), u - v \rangle \leq F_m(u) \leq F_m(v) + \langle \nabla F_m(v), u - v \rangle + \frac{H}{2} \|u - v\|_2^2.$$

**Assumption 2** (Lipschitzness). *Function $F_m : \mathcal{W} \to \mathbb{R}$ is $L$-Lipschitz, if for all $u, v \in \mathcal{W}$,*

$$|F_m(u) - F_m(v)| \leq L \|u - v\|_2.$$

Furthermore, the data distributions $\mathcal{D}_m$'s have to be "similar" so that all agents would benefit from collaboratively learning a single consensus model. Following Blum et al. (2017), we capture the similarity among the agents by assuming that a single model works well for all agents.

**Assumption 3** (Realizability). *There exists $w^\star \in \mathcal{W}$ such that $w^*$ is the shared minima for all agents, i.e., $F_m(w^*) = \min_{w \in \mathcal{W}} F_m(w)$ for all $m \in [M]$. We denote the set of shared minima by $\mathcal{W}^\star$. For simplicity, we assume that $F_m(w^\star) = 0$, for all $m \in [M]$.*

This assumption holds when using over-parameterized models that easily fit the combined training data on all the agents. Furthermore, when this assumption is not satisfied, it is unclear if returning a single consensus model, as usual in federated learning, is reasonable.

The learning goal is to output a model $w_R$ such that $F(w_R) \leq \epsilon$ for some fixed precision parameter $\epsilon > 0$. We denote the $\epsilon$-sub level sets of the function $F_m$ by $S_m^\star$, i.e., $S_m^\star = \{w \in \mathcal{W} | F_m(w) \leq \epsilon\}$. Then our realizability assumption implies that, $S^\star := \cap_{m \in [M]} S_m^\star \neq \emptyset$. The learning goal can be achieved if we output $w_R \in S^*$.

**Intermittently Communicating First-order (ICFO) Algorithms.** We focus on intermittently communicating first-order (ICFO) algorithms such as FEDAVG. More specifically, at each round $r \in [R]$, the server sends out the current model $w_{r-1}$ to all agents participating in the learning in this round. All participating agents $m$ query their first-order oracle [1] at the current model $\mathcal{O}_m(w_{r-1}) = (F_m(w_{r-1}), \nabla F_m(w_{r-1}))$ and send it to the server, which makes an update to the model. The detailed algorithm framework is described in Figure 4 and Algorithm 2 in the appendix.

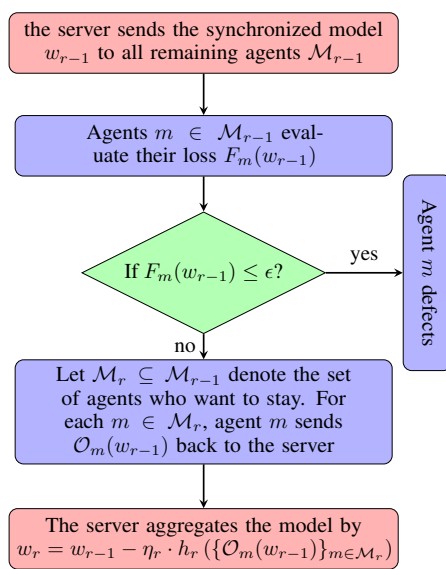

Figure 4: Round $r$ of ICFO algorithms.

Note that in the ICFO algorithm class, different algorithms can determine (i) the initialization $w_0$, (ii) step sizes $\eta_{1:R}$, and (iii) an aggregation rule $h_r(\cdot)$ for each round, which takes input $\{\mathcal{O}_m(w_{r-1})\}_{m \in \mathcal{M}_r}$ and outputs a vector in $Span\{\nabla F_m(w_{r-1})\}_{m \in \mathcal{M}_r}$. For example, FEDAVG will use uniform aggregation, i.e., set $h_r(\{\mathcal{O}_m(w_{r-1})\}_{m \in \mathcal{M}_r}) = \frac{1}{|\mathcal{M}_r|} \sum_{m \in \mathcal{M}_r} \nabla F_m(w_{r-1})$ for all $r \in [R]$. We say an ICFO algorithm is **convergent** if the algorithm will converge to optima when agents are irrational, i.e., when agents never drop out.

**Rational Agents.** Since agents have computation/communication costs to join this collaboration, they will defect when they are happy with the instantaneous model's performance on their local data. In particular, in the $r$-th round, after receiving a synchronized model $w_{r-1}$ from the server, agent $m$ will defect permanently if the current model $w_{r-1}$ is satisfactory, i.e., $F_m(w_{r-1}) \leq \epsilon$ for some fixed precision parameter $\epsilon > 0$. Then, agent $m$ will not participate in the remaining process of this iteration, including local training, communication, and aggregation. Thus, $m \notin \mathcal{M}_{r'}$ for $r' \geq r$, where $\mathcal{M}_{r'}$ is the set of participating agents in round $r'$. Note that a sequence of agents may defect for an algorithm $\mathcal{A}$, running over a set of rational agents. We say the defection behavior (of this sequence of agents) **hurts** (for algorithm $\mathcal{A}$) if the final output $w_R \notin S^\star$, i.e., we don't find a model in the $\epsilon$ sub-level set of an otherwise realizable problem.

In the field of game theory (and economics), the rationality of agents is commonly modeled as aiming to maximize their net utility, which is typically defined as the difference between their payoffs and associated costs, such as the difference between value and payment in auctions or revenue and cost in markets (Myerson, 1981; Huber et al., 2001; Börgers, 2015). In our model, each agent aims to obtain a loss below $\epsilon$ over their local data distribution (and any loss below $\epsilon$ is indifferent), and therefore, their payoff upon receiving a model $w$ is $v(w) = V \cdot \mathbb{I}[F_m(w) \leq \epsilon]$ for some $V > 0$. Their costs increase linearly with the number of rounds they participate in the training process. Therefore, the cost of participating in training for $r$ rounds is given by $c(r) = C \cdot r$ for some constant $C \ll V$. In this case, the optimal choice for the agent is to defect once they receive a model with a local loss below $\epsilon$ and never return in the future.

---

[1] In practical implementations of these algorithms, usually the agent sends back an unbiased estimate $(\widehat{F}_m(w_{r-1}), \nabla \widehat{F}_m(w_{r-1}))$ of $(F_m(w_{r-1}), \nabla F_m(w_{r-1}))$. This is attained by first sampling data-points $\{z_{r,k}^m \sim \mathcal{D}_m\}_{k \in K_r^m}$ on machine $m$ and returning $\left(\frac{1}{K_r^m} \sum_{k \in [K_r^m]} f(w_{r-1}; z_{r,k}^m), \frac{1}{K_r^m} \sum_{k \in [K_r^m]} \nabla f(w_{r-1}; z_{r,k}^m)\right)$. We focus on the setting with exact gradients and function values in our theoretical results but have stochasticity in our experiments.

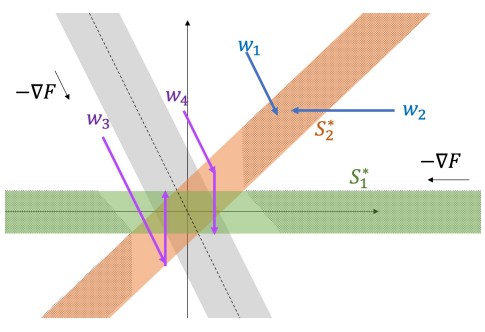 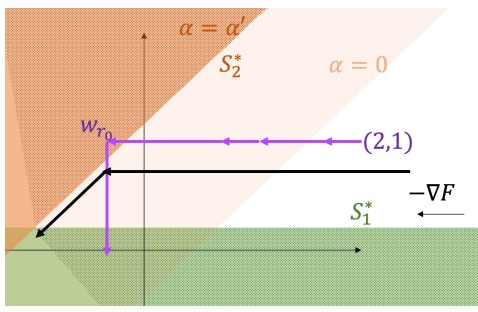

(a) Contour illustration of Example 1.          (b) Contour illustration of Observation 4.

Figure 5: Illustrations of the contours of the examples presented in Section 3.

## 3 WHEN DO DEFECTIONS HURT?

One might naturally wonder whether defections in an optimization process are universally detrimental, consistently harmless, or fall somewhere in between for any given algorithm. Our experiments, as shown in Figure 3, confirm the existence of harmful defections. However, it's worth noting that not all defections have a negative impact; some are actually benign.

**Observation 1.** *There exists a learning problem with two agents such that for any convergent ICFO algorithm $\mathcal{A}$, any defection will be benign, i.e., $\mathcal{A}$ will converge to a $w_R \in S^*$.*

Consider a 2-dimensional linear classification problem where each point $x \in \mathbb{R}^2$ is labeled by $w^* = \mathbb{1}[\mathbf{e}_1^\top x \leq 0]$. There are two agents where the marginal data distribution of agent 1 is uniform over the unit ball $\{x| \|x\|_2^2 \leq 1\}$ while agent 2's marginal data distribution is a uniform distribution over $\{\pm \mathbf{e}_1\}$. In this case, any linear model $w$ satisfying agent 1, i.e., $F_1(w) \leq \epsilon$ (for any appropriate loss function) will also satisfy agent 2. Therefore, for any convergent ICFO algorithm, if there is a defection, it must be either agent 2's defection or the defection of both agents. In either case, the defection is benign for any convergent ICFO algorithm. This example can be extended to a more general case where agent 1's $\epsilon$-sub level set is a subset of agent 2, i.e., $S_1^* \subset S_2^*$.

In practice, the ideal scenario where defections are benign is uncommon, as most defections tend to have a negative impact (refer to Figure 1 for an illustration). Thus, the extent to which defections are harmful is largely algorithm-dependent. To gain insights into which algorithms can mitigate the negative effects of defections, we will explore the roles of initialization, step size, and aggregation methods in this section. These insights will inform our algorithmic design in the subsequent section. Let's consider the following example for further clarity.

**Example 1.** *Consider an example with two agents with $\mathcal{W} = \mathbb{R}^2$, $F_1(w) = |(0,1)^\top w|$ and $F_2(w) = |(1,-1)^\top w|$ as illustrated in Figure 5a. Then we have the optimal model $w^* = (0,0)$ and the subgradients $\nabla F_1(w) = \text{sign}((0,1)^\top w) \cdot (0,1)$ and $\nabla F_2(w) = \text{sign}((1,-1)^\top w) \cdot (1,-1)$. Note that the functions can be smoothed, and the following observations remain.*

**Observation 2.** *There exists an initialization such that harmful defections are inevitable for any ICFO algorithm in Example 1. Specifically, there exists a $w_0 \in S_1^* \cup S_2^*$ such that for any ICFO algorithm $\mathcal{A}$, when initialized at $w_0$, $\mathcal{A}$ will converge to a model $w_R \notin S^*$. Furthermore, there exists a $w_0 \in S_1^* \cup S_2^*$ such that any ICFO algorithm initialized at $w_0$ will converge to a $w_R$ with $F(w_R) \geq 1/2 > 0 = F(w^*)$.*

For example, if the algorithm is initialized at $w_0 = (1,1)$, agent 2 defects given $w_0$ and an ICFO algorithm can only update in the direction of $\nabla F_1(w) = (0, \text{sign}((0,1)^\top w))$. It will converge to a $w_R \in \{(1,\beta)|\beta \in \mathbb{R}\}$ and incurs $F(w_R) \geq \frac{1}{2}$ since $F(w) \geq \frac{1}{2}$ for all $w \in \{(1,\beta)|\beta \in \mathbb{R}\}$. Note that if the algorithm is initialized at any point in the dotted region of Figure 5a, which is a subset of $S_1^* \cup S_2^* \setminus S^*$, one agent would defect immediately. The algorithm can only make updates according to the remaining agent and leads to a $w_R \notin S^*$. We refer to the dotted region as the *bad region*. No ICFO algorithm can avoid harmful defections when initialized in the bad region. The next question is: can specific algorithms avoid harmful defections when initialized in the good region? We then

notice that we must be very careful about step sizes. We need to fix an aggregation method to discuss the effects of step sizes, and here we consider uniform aggregation, for example.

**Observation 3.** *Consider randomly choosing the initialization $w_0$ from $\{w| \|w\|_2 \leq 1\}$. In Example 1, with probability $1 - 2\epsilon$ over $w_0$, there exists a sequence of step sizes $\eta_{1:R}$ such that the ICFO algorithm with uniform aggregation cannot avoid harmful defections, i.e., $w_R \notin S^*$.*

In Figure 5a, if we initialize at any point in the white area, we can set the step size s.t. the algorithm will converge to a $w_R \notin S^*$. For example, if we initialize at $w_1$ or $w_2$ in Figure 5a, the algorithm will step into the bad initialization region after the update by setting step size according to the blue trajectories. If we initialize at $w_3$ or $w_4$ in Figure 5a, then by appropriately setting the step sizes, the algorithm will follow the purple trajectory in the figure. In this case, the algorithm will step into $S_2^*$ after round 1, followed by agent 2 defecting, and eventually converge to a $w_R \in S_1^* \setminus S_2^*$. Observation 3 can be extended to other aggregation methods. If we initialize at $w_2$, except making an update in the direction towards $(2\epsilon, \epsilon)$ (which requires more than local information), there is a choice of step size s.t. the algorithm will step into the bad region. However, having a good initialization and carefully selected step sizes will not be sufficient. Any ICFO algorithm with uniform aggregation cannot avoid harmful defections.

**Observation 4.** *For any ICFO algorithm $\mathcal{A}$ with uniform aggregation, there exists a learning problem of two agents in which $\mathcal{A}$ initialized in the good region will converge to a model $w_R \notin S^*$.*

We construct a slightly different example from Example 1. As illustrated in Figure 5b, we consider a set of problems $\{F_1(w) = \max((0,1)^\top w, 0), F_2(w) = \max((1,-1)^\top w + \alpha, 0)|\alpha \geq 0\}$. For all $\alpha \geq 0$ illustrated in Figure 5b, the problem characterized by $\alpha$ is denoted by $P_\alpha$. Suppose we initialize at a $w_0$ which is not in the union of $\epsilon$-sub level sets $S_1^* \cup S_2^*$ of $P_\alpha$ for all $\alpha$. Otherwise, the initialization will lie in the bad region of $P_\alpha$ for some $\alpha$. W.l.o.g., suppose the algorithm initializes at $w_0 = (2, 1)$. When no agents defect, the average gradient $\nabla F(w) = (1/2, 0)$ is a constant, and any ICFO algorithm can only move in the direction of $(-1, 0)$ until agent 2 defects. Among all choices of $\alpha$, agent 2 will defect earliest in the problem $P_0$ (since any model good for agent 2 for some $\alpha > 0$ is also good for agent 2 for $\alpha = 0$). The algorithm will follow the purple trajectory in Figure 5b. Before agent 2's defection, the algorithm's updates are identical for all $\alpha \geq 0$. Denote by $r_0$ the round at which agent 2 is defecting in $P_0$. Then there exists an $\alpha'$ (illustrated in the figure) s.t. $w_{r_0}$ lies in the bad region of problem $P_{\alpha'}$. Therefore, in problem $P_{\alpha'}$, the algorithm will output a $w_R \notin S^*$. Especially, we have $F(w_R) = 1/2 > 0 = F(w^*)$.

Note that we can avoid defections by non-uniform aggregation. For example, when the algorithm reaches a $w$ with low $F_2(w)$ and high $F_1(w)$, we can project $\nabla F_1(w)$ into the orthogonal space of $\nabla F_2(w)$ and update in the projected direction. Figure 5b provides a visual representation of the effectiveness of the projection technique, which is shown by the black trajectory within the context of this example. This not only showcases the benefits of projection but also serves as a motivation behind the development of our algorithm.

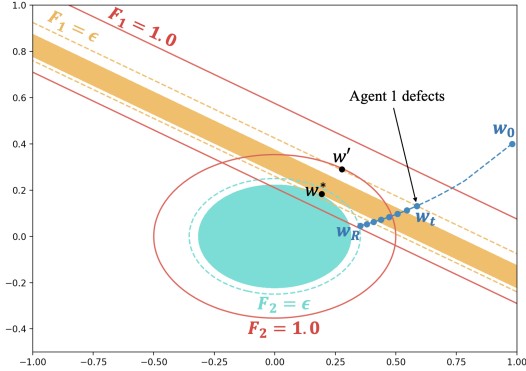

Figure 6: In this illustration, agent 1's loss function $F_1$ is piece-wise linear, while agent 2's loss function $F_2$ is quadratic. Both functions are truncated to have a minimum value of zero and smoothed. The figure highlights zero-loss regions: filled orange for $F_1$ and green for $F_2$. Their shared optimum is $w^\star$ where $F_m(w^\star) = 0$. Assumptions 1, 2, and 3 are satisfied, but Assumption 4 is not, as the gradients $\nabla F_1(w')$ and $\nabla F_2(w')$ are parallel at $w'$. Dashed contour lines indicate the $\epsilon$ level set ($F_m(w) = \epsilon$), and solid lines represent the 1 level set ($F_m(w) = 1$). Starting from $w_0$, any trajectory must pass the orange region to reach the $\epsilon$-sub level set of $F_2$. When the model updates to $w_t$ at time $t$, agent 1 would defect as $F_1(w_t) \leq \epsilon$. Subsequent updates follow $\nabla F_2(\cdot)$, and with a small step size, the model converges to $w_R$. This final model deviates from agent 1's $\epsilon$ level set, resulting in a poor performance with $F_1(w_R) = 1$.

---

**Algorithm 1:** Adaptive Defection-aware Aggregation for Gradient Descent (ADA-GD)

---

1 **Parameters:** step size $\eta$, initialization $w_0$
2 **for** $t = 1, 2, \ldots$ **do**
3     Compute $\nabla F_m(w_{t-1})$ for all $m \in [M]$
4     **Predict defecting agents:** $D = \{m \in [M] : F_m(w_{t-1}) - \eta \|\nabla F_m(w_{t-1})\|_2 \leq 2\epsilon\}$
5     **Predict non-defecting agents:** $ND = [M] \setminus D$
6     **if** *Both $D$ and $ND$ are non-empty* **then**
7        Compute $\nabla F_{ND}(w_{t-1}) = \sum_{m \in ND} \nabla F_m(w_{t-1})$
8        Let $P = Span\{\nabla F_n(w_{t-1}) : n \in D\}^{\perp}$
9        Project $\nabla F_{ND}(w_{t-1})$ and normalize $\nabla \widetilde{F}_{ND}(w_{t-1}) = \frac{\Pi_P(\nabla F_{ND}(w_{t-1}))}{\|\Pi_P(\nabla F_{ND}(w_{t-1}))\|_2}$
10        $g_t = -\min\left\{\|\Pi_P(\nabla F_{ND}(w_{t-1}))\|_2, 1\right\} \cdot \nabla \widetilde{F}_{ND}(w_{t-1})$      ▷ Case 1
11     **if** *$D$ is empty* **then**
12        $g_t = -\min\{\|\nabla F(w_{t-1})\|_2, 1\} \cdot \frac{\nabla F(w_{t-1})}{\|\nabla F(w_{t-1})\|_2}$      ▷ Case 2
13     **if** *$ND$ is empty* **then**
14        **return** $\widehat{w} = w_{t-1}$      ▷ Case 3
15     $w_t = w_{t-1} + \eta g_t$

---

## 4 DISINCENTIVIZING DEFECTIONS THROUGH A DIFFERENT AGGREGATION METHOD

In this section, we will introduce a new method. Before we state our algorithm, let's recall the definition of the orthogonal complement of a subspace.

**Definition 1** (Orthogonal Complement). *Let $V$ be a vector space with an inner-product $\langle \cdot, \cdot \rangle$. For any subspace $U$ of $V$, we define the orthogonal complement as $U^{\perp} := \{v \in V : \langle v, u \rangle = 0, \ \forall u \in U\}$. We also define the projection operator onto a subspace $U$ by $\Pi_U : V \to U$.*

Our algorithm ADA-GD is stated in Algorithm 1. At our method's core is an adaptive aggregation approach for the gradients received from agents, which disincentivizes participating agents' defection during the training. However, disincentivizing defection is impossible if two agents are very "similar". Consider the example of $F_1(w) = \|w\|_2^2$ and $F_2 = \epsilon \|w\|_2^2$. Whenever we reach some $w_t$ with $\|w_t\|_2^2 \in (\epsilon, 1]$ (otherwise, we have to jump from some $w_{t-1}$ with $\|w_{t-1}\|_2^2 > 1$ to some $w_t$ with $\|w_t\|_2^2 \leq \epsilon$), agent 2 will defect. Therefore, we must introduce some heterogeneity assumption implying that the functions are *"dissimilar enough"*.

**Assumption 4** (Minimal Heterogeneity). *Consider differentiable functions $\{F_m : \mathbb{R}^d \to \mathbb{R}\}$. Then for all $w \in \mathcal{W} \setminus \mathcal{W}^{\star}$, all non-zero vectors in $\{\nabla F_m(w) : m \in [M]\}$ are linearly independent.*

**Remark.** *To justify the necessity of this linear independence assumption, one may argue that in the above simple example, the defection is benign. We provide a more complicated example where linear dependence leads to harmful defections in Fig 6.*

Under Assumptions 1, 2, 3, 4, we show that our algorithm is legal (i.e., any denominator in Algorithm 1 is non-zero) and converges to an approximately optimal model.

**Theorem 1.** *Suppose $\{F_m\}_{m \in [M]}$, satisfy Assumptions 1, 2, 3, 4. If we choose $w_0 \notin \cup_{m \in [M]} S_m^{\star}$ and $\eta \leq \min\left\{\frac{\epsilon}{L}, \sqrt{\frac{\epsilon}{2H}}, \sqrt{\frac{2}{H}}, \frac{1}{MH}\right\}$, then no agents will defect and Algorithm 1 will output $\widehat{w}$ that satisfies $F(\widehat{w}) \leq 4\epsilon$.*

Now we outline the high-level ideas behind Algorithm 1. Intuitively, an agent is close to defection if a direction $u$ exists such that she would defect after receiving $w_{t-1} - \eta u$. Hence, the server has to carefully tune the update direction to avoid the defection of this agent.

Suppose no agent has defected upon receiving $w_{t-1}$. Then, the server will receive the first-order information $\mathcal{O}_m(w_{t-1})$ from all agents and determine a direction to update $w_{t-1}$ to $w_t$. In Algorithm 1, the server first predicts which agents are close to defection through a linear approximation

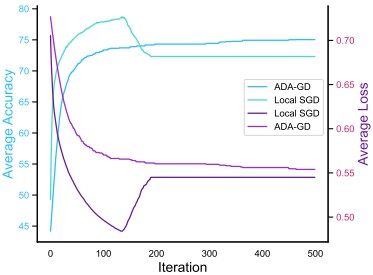 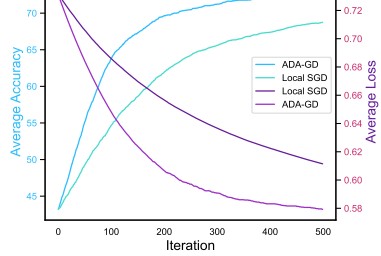

(a) Both algorithms use $\eta = 0.001$ and $\epsilon = 0.4$.  (b) $\epsilon = 0.5$, $\eta_{\text{LSGD}} = 2e^{-5}$, $\eta_{\text{ADA-GD}} = 2e^{-4}$.

Figure 7: We study the performance of our method and federated averaging (local SGD) for binary classification on two classes of CIFAR-10 (Krizhevsky et al., 2009) with two agents. The data heterogeneity $q = 0.9$ and local update step $K = 1$ for both experiments. Figure 7a shows that Local SGD results in agent defection around the 130-th iteration, leading to a considerable decay in the accuracy of the final model. In sharp contrast, our proposed method effectively eliminates defection, enabling the continuous improvement of the model throughout the training process. Remark that it's unfair to compare the highest accuracy of Local SGD with our method as Local SGD is not defection aware and could not simply stop at the highest point. In Figure 7b, we study the performance of both algorithms with the largest step size such that no agents will defect. We observe that ADA-GD can avoid defection even when employing a considerably larger step size than Local SGD. As a result, this enables us to attain a significantly improved final model. These results support our theoretical findings and validate the effectiveness of our method.

(see line 4) and calls them 'defecting' agents (although these agents might not defect because of the slack in the precision in line 4). If all agents are 'defecting' (case 3), the server will output the current model $w_{t-1}$ as the final model (see line 14). The output model is approximately optimal since each agent has a small $F_m(w_{t-1})$ when they are 'defecting'. If no agent is 'defecting' (case 2), the server is certain that all agents will not defect after the update and thus will update in the steepest descending direction, i.e., the gradient of the average loss $\nabla F(w_{t-1})$ (see line 12). If there exist both 'defecting' and 'non-defecting' agents (case 1), the server will aggregate the gradients from non-defecting agents and project them to the orthogonal complement of the space spanned by the gradients of the 'defecting' agents to guarantee that 'defecting' agents will not defect. Then, the server will update the current model using the normalized projected gradients (see line 10). By induction, we can show that no agents will defect. Furthermore, we can prove that we will make positive progress in decreasing the average loss at each round, and as a result, our algorithm will ultimately converge to an approximately optimal model. Our analysis is inherently case-based, making it difficult to replicate a simple distributed gradient descent analysis that proceeds by showing that we move in a descent direction at each iteration. Consequently, offering a convergence rate poses a significant challenge and remains an open question.

Note that Algorithm 1 fits in the class of ICFO algorithms specified in Section 2, as orthogonalization and normalization return an output in the linear span of the machines' gradients. In Figure 7, we compare ADA-GD against FEDAVG, demonstrating the benefit of adaptive aggregation.

## 5 DISCUSSION

In this work, we initiate the study of defections in federated learning. Defections are an unavoidable part of federated learning and can have drastic consequences for the performance/robustness of the final model. We theoretically and empirically characterize the effects of defection and demarcate between benign and harmful defections. We underline the importance of adaptive aggregation in avoiding defections and propose an algorithm ADA-GD with a provable guarantee and a promising empirical performance. There are several open questions and avenues for future work. Firstly, all of the theoretical analyses in this paper assume access to the exact first-order oracles as opposed to stochastic oracles. It is unclear how our Algorithm 1 can incorporate local steps. Local steps complicate the defection model, as agents might not even wait until communication happens to defect.

This makes the orthogonalization trick in Algorithm 1 challenging to apply. We further discuss this and other future directions for our work in appendix E.

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

## A  MISSING DETAILS FROM SECTION 2

---
**Algorithm 2:** ICFO algorithms

---
1 **Parameters:** Intialization $w_0$, step size $\eta_{1:R}$, aggregation rules $\{h_r(\cdot)\}_{r=1}^R$, $\mathcal{M}_0 = [M]$
2 **for** $r = 1, \ldots, R$ **do**
3     The server sends out $w_{r-1}$ to agents in $\mathcal{M}_{r-1}$
4     For each agent, $m \in \mathcal{M}_{r-1}$, $m$ can decides to drop out or to stay. Let $\mathcal{M}_r \subseteq \mathcal{M}_{r-1}$ denote the set of agents who want to stay
5     For each $m \in \mathcal{M}_r$, agent $m$ sends $\mathcal{O}_m(w_{r-1})$ back to the server
6     The server aggregates the model by $w_r = w_{r-1} - \eta_r \cdot h_r\left(\{\mathcal{O}_m(w_{r-1})\}_{m \in \mathcal{M}_r}\right)$
7 **return** $\widehat{w} = w_R$

---

---
**Algorithm 3:** FEDAVG w/o any defections

---
1 **Input:** step size $\eta$
2 **Initialize** $w_0^m = w_0 = 0$ on all agents $m \in [M]$
3 **for** $r \in \{1, \ldots, R\}$ **do**
4     **for** $m \in [M]$ ***in parallel*** **do**
5        Set $w_{r,1}^m = w_{r-1}$
6        **for** $k \in \{1, \ldots, K\}$ **do**
7           Sample $z_{r,k}^m \sim \mathcal{D}_m$
8           Compute stochastic gradient at $w_{r,k}^m$, $g_{r,k}^m \leftarrow \nabla f(w_{r,k}^m; z_{r,k}^m)$
9           Update $w_{r,k+1}^m \leftarrow w_{r,k}^m - \frac{\eta}{K_r^m} g_{r,k}^m$
10        **Communicate to server:** $w_r^m \leftarrow w_{r,K_r^m+1}^m = w_{r-1} - \frac{\eta}{K_r^m}\sum_{k \in [K_r^m]} g_{r,k}^m$
11     **Communicate to agents:** $w_r \leftarrow \frac{1}{M}\sum_{m \in [M]} \cdot w_r^m$
12 **Output:** Return $w_R$

---

## B  MORE RELATED WORK

Designing incentive mechanisms for federated learning has recently received much attention, but it remains hard to solve. The main difficulty lies in an *"information asymmetry"* (Kang et al., 2019). The server does not know the available computation resources and the data sizes on the devices for training. Furthermore, it doesn't know the local data quality of a device and can't estimate it using common metrics such as data Shapley (Ghorbani & Zou, 2019) as it doesn't have access to raw data. As a result, the server may incur a high cost when providing incentives to the devices to encourage truthfulness or avoid defections. Or worse, it might not even be able to incentivize desired behavior.

In this section, we survey some recent advances in solving this problem. We divide the survey into four parts,

1. In section B.1, we survey papers that tackle the problem of *agent selection* based on several factors, but primarily the value of the data the agent can provide. This is an important step because for cross-device FL to work, the agents defining problem (2) should (i) represent the meta distribution $\mathcal{P}$ well, (ii) have enough data samples/computational power to converge to a solution for (2).

2. In section B.2, we survey the papers that assume the pool of agents is fixed, but the agents are self-serving and want to contribute the least amount of data possible. This is because data collection and privacy costs are involved with sharing data. The works in this category also tackle the "free-rider" problem (Karimireddy et al., 2022) so a handful of devices don't collect most of the data at equilibrium.

3. In section B.3, we survey papers relevant to a simpler problem called *cross-silo FL* (Kairouz et al., 2019) where $\mathcal{P}$ is supported on a finite set of machines, and as a result, we can directly consider problem (2). Defection is not a problem in this setting, and we are more interested in incentivizing the machines to contribute higher-quality model updates.

4. Finally, in section B.4, we survey the very limited work in the general problem we are interested in solving, i.e., avoiding defections in cross-device FL.

There is another orthogonal line of research on fairness in FL (see Ezzeldin et al. (2021) and the references within, for example).

## B.1 DATA VALUATION AND AGENT SELECTION

There are two main series of works that are most relevant. The first one studied the design of incentive mechanisms under a principal-agent model using contract theory.

Kang et al. (2019) proposed a contract theory-based incentive mechanism for federated learning in mobile networks. They considered the principal-agent model that there are $M$ types of agents, where the agent's local data quality defines the type. The server does not know the type of each device and only has a prior distribution over all agent types. It aims to design an incentive-compatible mechanism that incentivizes every agent to behave truthfully while maximizing the server's expected reward. Tian et al. (2021) extended this model by considering the agent's willingness to participate as part of the type. As a result, they proposed a two-dimensional contract model for the incentive mechanism design. Cong et al. (2020) considered a similar model where the data quality and cost define the agent's type. The server decides payment and data acceptance rates based on the reported agent type. The authors proposed implementing the payment and data acceptance rate functions with neural networks and showed they satisfy proper properties such as incentive compatibility. Another related work is by Zeng et al. (2020), who proposed a multi-dimension auction mechanism with multiple winners for agent selection. The authors presented a first-price auction mechanism and analyzed the server's equilibrium state.

Besides applying contract theory to incentivize agents' truthful behavior, there has also been another series of work that focuses on designing reputation-based agent selection schemes to select high-quality agents. Specifically, Kang et al. (2019) proposed a model that works as follows: the server computes the reputation of every agent and selects agents for the federated learning based on the reputation scores, and the reputation of every agent is updated by a management system after finishing an FL task, where a consortium blockchain implements the reputation management system. Zhang et al. (2021) extended the reputation-based agent selection scheme with a reverse auction mechanism to select agents by combining their bids and comprehensive reputations. Xu & Lyu (2020) proposed a different reputation mechanism that computes the reputation of every agent based on the correlation between their report and the average report. Each agent will be rewarded according to their reputation, and those with low reputations will be removed.

Finally, it is also worth mentioning two other related works. Richardson et al. (2020) designed another data valuation method. They collect an independent validation data set and value data through an influence function, defined as the loss decreases over the validation data set when the data is left out. And Hu & Gong (2020) cast the problem as a Stackelberg game and choose a subset of agents, which they claim are most likely to provide high-quality data. They provide extensive experiments to test their idea.

## B.2 INCENTIVIZING AGENT PARTICIPATION AND DATA-COLLECTION

Usually, the utility of participating agents is a function of model accuracy, computation/sample collection cost, and additional constant cost, e.g., communication cost and payment to the server. In a simple case where the model accuracy is a function of the total number of input samples, Karimireddy et al. (2022); Zhang et al. (2022) show that when the server gives the same global model to every agent like federated averaging (McMahan et al., 2016a) does, a Nash Equilibrium exists for individual sample contributions. But, at this Nash Equilibrium, agents with high sampling costs will be "free-riders" and will not contribute any data, while agents with low sampling costs will contribute most of the total data.

To alleviate this situation, Karimireddy et al. (2022) propose a mechanism to incentivize agents to contribute more by customizing the final model's accuracy for every agent, which means sending different models to different agents. They show that their mechanism is "data-maximizing" in the face of rational agents. Sim et al. (2020) study a similar setting as Karimireddy et al. (2022) and aim to provide model-based rewards. Specifically, the authors use information gain as a metric for data

valuation and show their reward scheme satisfies some desirable properties. Zhang et al. (2022) also study an infinitely repeated game, where the utility is the discounted cumulative utility, and propose a cooperative strategy to achieve the minimum number of free riders.

Zhan et al. (2020) formulate the problem as a Stackelberg game, where the server decides the payment amount first, and agents decide the amount of data they are willing to contribute. The server's utility is the model accuracy minus the payment, while the agents' utility is the payment (proportional to the contributing data size) minus the computation cost. In this setting, the agents do not care about accuracy, which differs from our case.

In a bit of orthogonal work, Cho et al. (2022) propose changing objective (2) itself to maximize the number of agents that benefit from collaboration. They provide some preliminary theoretical guarantees for a simple mean estimation problem. While in our setting, we assume the server can get the model from agents. Liu & Wei (2020) study the problem of eliciting truthful reports of the learned model from the agents by designing proper scoring rules. Specifically, the authors consider two settings where the server has a ground truth verification data set or only has access to features. The authors demonstrate the connections between this question and proper scoring rule and peer predictions (i.e., information elicitation without verification) and test the performance with real-world data sets.

### B.3    MAXIMIZING DATA QUALITY IN CROSS-SILO FL

Kairouz et al. (2019) demarcate between two prominent federated learning paradigms: cross-silo and cross-device. We have already discussed an example of cross-device Fl: training on mobile devices. Cross-silo FL captures the traditional training in data centers or between big organizations with similar interests. One example is a collaboration between medical institutions to improve their models without leaking sensitive patient information Bergen & Petryshen (2012). In cross-silo FL, usually, the agents initiate the FL process and pay the central server for global aggregation. As a result, defection is not an issue in cross-silo FL because, ultimately, the goal is to develop better individual models. Unfortunately, there can still be "free-riding" behavior in the cross-silo setting (Zhang et al., 2022; Richardson et al., 2020) as the devices have incentives to contribute less to maximize their own benefit. Therefore, maximizing the data quality is one of the main problems in cross-silo FL.

Xu et al. (2021) propose a heuristic Shapley score based on the gradient information from each agent. The score is calculated after communication by comparing the alignment of an agent's gradient with the aggregate gradient. Then the agents with a high score are provided an un-tarnished version of the aggregated gradient, while the agents with a lower score only get a noisy version of the gradient. This incentivizes devices to provide higher-quality gradient updates to get a final model close to the model of the server. There are, unfortunately, no guarantees showing this won't hurt the optimization of objective (2), or it at least provably maximizes data quality in any sense. A similar idea has also been used in Shi et al. (2022). Zheng et al. (2021) also propose an auction mechanism modeled using a neural network that decides the appropriate perturbation rule for agents' gradients and an aggregation rule that helps recover a good final guarantee despite this perturbation. Richardson et al. (2020) take a different approach, and instead of perturbing the model updates, they make budget-bound monetary payments to devices. Their metric is like the leave-one-out metric in data valuation but for model updates. Finally, Tang & Wong (2021) formulate a social welfare maximization problem for cross-silo FL and propose an incentive mechanism with preliminary theoretical guarantees.

### B.4    TOWARDS AVOIDING DEFECTIONS IN CROSS-DEVICE FL

There doesn't exist any theoretical work for our proposed problem, i.e., avoiding agent defection while optimizing problem (2) using an iterative algorithm with several communication rounds. There are, however, some empirical insights in other works.

The most relevant work is about MW-FED algorithm (Blum et al., 2021), an algorithm that fits into the intermittent communication model and explicitly slows down the progress of devices closer to their target accuracies. Specifically, the algorithm asks the devices to report their target accuracies at the beginning of training. Then after each communication round, the devices report their validation accuracy on the current model. The server then uses a multiplicative weight update rule to devise a

sample load for each device for that communication round. Intuitively, devices closer to their target accuracies get a lighter sample load and vice-versa. Practically this ensures that the devices are all satisfied at roughly the same point, thus avoiding any incentives to defect. While MW-FED hasn't been analyzed in the context of federated learning, it is well-known that it has an optimal sample complexity for optimizing distributed learning problems where the goal is to come up with a single best model for all agents, much like problem (2).

## C   MISSING EXAMPLES AND PROOFS FROM SECTION 4

### C.1   PROOF OF THEOREM 1

*Proof of Theorem 1.* We will show that algorithm 1 with a small enough step-size sequence,

1. is legal, i.e., at time $t$, if we are in case 1, we have $\|\Pi_P\left(\nabla F_{ND}(w_{t-1})\right)\|_2 \neq 0$; if we are in case 2, we have $\|\nabla F(w_{t-1})\|_2 \neq 0$.

2. will not cause any agent to defect, and

3. will terminate and output a model $\widehat{w}$ which is approximately optimal.

For the first property, we have the following lemma.

**Lemma 1.** *Under the same conditions of Theorem 1. Suppose the algorithm is in case 1 or case 2 at any time step $t$. If we are in case 1, we have $\|\Pi_P\left(\nabla F_{ND}(w_{t-1})\right)\|_2 \neq 0$; if we are in case 2, we have $\|\nabla F(w_{t-1})\|_2 \neq 0$.*

Now to see the second property, note that defections can only happen if (i) the algorithm runs into case 1 or 2, (ii) it makes the corresponding update, and (iii) the agent chooses to defect after seeing the updated model.

**Lemma 2** (No agent will defect in case 1 and case 2)**.** *Under the same conditions as in Theorem 1. Suppose the algorithm is in case 1 or case 2 at any time step $t$, and no agent has defected up to time step $t$. If $\eta \leq \sqrt{\epsilon/(2H)}$, no defection will occur once the update is made.*

The reason that no agent defects in case 1 is that we create our update direction in case 1 so that for all agents in $D$, the update is orthogonal to their current gradient, and thus they don't reduce their objective value. And for all agents in $ND$, they do make progress, but we control the step size so that they don't make *"too much progress"*. Thus no agent defects in case 1. Similarly, in case 2 we avoid defections by ensuring the step size is small enough.

Finally, we show that the algorithm will terminate and the returned model returned is good.

**Lemma 3** (The algorithm will terminate)**.** *Under the same conditions of Theorem 1, Algorithm 1 terminates.*

To prove that the algorithm terminates, we first show that the algorithm makes non-zero progress on the average objective every time it is in case 1 and case 2 (which is shown in lemma 7), which implies that the average loss will converge. Then we prove that the average loss can only converge to zero, and thus, we will certainly get into case 3.

**Lemma 4** (The returned model is good)**.** *Under the same conditions as in Theorem 1. Suppose Algorithm 1 terminates in case 3 at time step $t$, and no agent has defected up to time step $t$. If $\eta \leq \min\left\{\sqrt{2\epsilon/H}, \epsilon/L\right\}$, then the algorithm will output $\widehat{w}$ that satisfies $F(\widehat{w}) \leq 4\epsilon$.*

Thus combining the three properties, we can conclude that the algorithm outputs a good model and avoids defections. This finishes the proof. $\qquad\square$

## C.2   PROOF OF LEMMA 1

First, consider **case 2**. Suppose that $\|\nabla F(w_{t-1})\|_2 = 0$. If there exists $n \in [M]$ s.t. $F_n(w_{t-1}) > F_n(w^*) = 0$, then we have

$$F(w^\star) \geq F(w_{t-1}),$$
$$= \frac{1}{M} \sum_{m \in [M]} F_m(w_{t-1}),$$
$$\geq \frac{1}{M} \sum_{m \neq n} F_m(w^\star) + \frac{F_n(w_{t-1})}{M},$$
$$> \frac{1}{M} \sum_{m \neq n} F_m(w^\star) + \frac{F_n(w^\star)}{M},$$
$$= F(w^\star),$$

which is a contradiction. Hence we have $F_n(w_{t-1}) = 0$ for all $n \in [M]$, which contradicts with the condition that $D$ is empty due to line 4 of Algorithm 1.

Next, consider **case 1**. We first introduce the following lemma.

**Lemma 5.** *Suppose Assumption 4 holds. For all $A \subset [M]$ and $B = [M] \setminus A$, if $F_A(\cdot) := \sum_{m \in A} F_m(\cdot)$, then for all $w \in \mathcal{W}$, such that $\nabla F_A(w) \neq 0$, $\nabla F_A(w) \notin Span\{\nabla F_m(w) : m \in B\}$.*

By combining Assumption 4 and lemma 5, we know that $\|\Pi_P(\nabla F_{ND}(w_{t-1}))\|_2 = 0$ implies that $\|\nabla F_{ND}(w_{t-1})\|_2 = 0$. This implies that $F_{ND}(w_{t-1}) = F_{ND}(w^\star) = 0$ for $w^\star \in \mathcal{W}^\star$, which contradicts with the definition of ND (see line 5 of Algorithm 1).

## C.3   PROOF OF LEMMA 2

*Proof.* We begin with introducing the following lemma.

**Lemma 6** (Predicting defections with single first-order oracle call)**.** *Under the same conditions of Theorem 1, at any time step $t$ assuming $\eta \leq \sqrt{2/H}$,*

- *if agent $m \in D$ then $F_m(w_{t-1} - \eta \nabla \widetilde{F}_m(w_{t-1})) \leq 3\epsilon$, and*

- *if agent $m \in ND$ then $F_m(w_{t-1} - \eta \nabla \widetilde{F}_m(w_{t-1})) > 2\epsilon$,*

*where $\nabla \widetilde{F}_m(w_{t-1}) = \frac{\nabla F_m(w_{t-1})}{\|\nabla F_m(w_{t-1})\|_2}$ is the normalized gradient.*

Now we are ready to prove lemma 2. First, assume we are in case 1 at time $t$. Let's first show that we don't make any agent $m \in D$ defect,

$$F_m(w_t) \geq F_m(w_{t-1}) + \langle \nabla F_m(w_{t-1}), g_t \rangle,$$
$$= F_m(w_{t-1}),$$
$$> \epsilon,$$

as $g_t \perp \nabla F_m(w_{t-1})$ for all $m \in D$ by design and no agent defected up to time $t$. For any non-defecting agent $m \in ND$ we have,

$$F_m(w_{t-1} + \eta_t g_t) - F_m(w_{t-1} - \eta_t \nabla \widetilde{F}_m(w_{t-1}))$$
$$\geq^{\text{(convexity)}} \left\langle \nabla F_m(w_{t-1} - \eta_t \nabla \widetilde{F}_m(w_{t-1})), \eta_t(g_t + \nabla \widetilde{F}_m(w_{t-1})) \right\rangle,$$
$$= \left\langle \nabla F_m(w_{t-1} - \eta_t \nabla \widetilde{F}_m(w_{t-1})) - \nabla F_m(w_{t-1}) + \nabla F_m(w_{t-1}), \eta_t(g_t + \nabla \widetilde{F}_m(w_{t-1})) \right\rangle,$$
$$\geq^{\text{(C.S. inequality)}} \left\langle \nabla F_m(w_{t-1}), \eta_t(g_t + \nabla \widetilde{F}_m(w_{t-1})) \right\rangle$$
$$- \left\| \nabla F_m(w_{t-1}) - \nabla F_m(w_{t-1} - \eta_t \nabla \widetilde{F}_m(w_{t-1})) \right\|_2 \cdot \left\| \eta_t(g_t + \nabla \widetilde{F}_m(w_{t-1})) \right\|_2,$$

$$\geq^{\text{(Ass. 1)}} \left\langle \nabla F_m(w_{t-1}), \eta_t(g_t + \nabla \widetilde{F}_m(w_{t-1})) \right\rangle - \eta_t^2 H \left\| \nabla \widetilde{F}_m(w_{t-1}) \right\|_2 \cdot \left\| g_t + \nabla \widetilde{F}_m(w_{t-1}) \right\|_2,$$

$$\geq^{\text{(normalized gradients)}} \left\langle \nabla F_m(w_{t-1}), \eta_t(g_t + \nabla \widetilde{F}_m(w_{t-1})) \right\rangle - 2\eta_t^2 H,$$

$$= \eta_t \left\| \nabla F_m(w_{t-1}) \right\|_2 \left( 1 + \left\langle \nabla \widetilde{F}_m(w_{t-1}), g_t \right\rangle \right) - 2\eta_t^2 H,$$

$$\geq -2\eta_t^2 H,$$

Re-arranging this gives the following,

$$F_m(w_{t-1} + \eta_t g_t) \geq F_m(w_{t-1} - \eta_t \nabla \widetilde{F}_m(w_{t-1})) - 2\eta_t^2 H$$
$$>^{\text{(lemma 6)}} 2\epsilon - 2\eta_t^2 H,$$
$$\geq \epsilon,$$

where we assume that $\eta_t \leq \sqrt{\frac{\epsilon}{2H}}$.

Now assume instead we are in case 2 at time $t$. For agent $m \in [M]$ we have,

$$F_m(w_{t-1} + \eta_t g_t) - F_m(w_{t-1} - \eta_t \nabla \widetilde{F}_m(w_{t-1}))$$

$$\geq^{\text{(convexity)}} \left\langle \nabla F_m(w_{t-1} - \eta_t \nabla \widetilde{F}_m(w_{t-1})), \eta_t(g_t + \nabla \widetilde{F}_m(w_{t-1})) \right\rangle,$$

$$= \left\langle \nabla F_m(w_{t-1} - \eta_t \nabla \widetilde{F}_m(w_{t-1})) - \nabla F_m(w_{t-1}) + \nabla F_m(w_{t-1}), \eta_t(g_t + \nabla \widetilde{F}_m(w_{t-1})) \right\rangle,$$

$$\geq^{\text{(C.S. inequality)}} \left\langle \nabla F_m(w_{t-1}), \eta_t(g_t + \nabla \widetilde{F}_m(w_{t-1})) \right\rangle$$
$$- \left\| \nabla F_m(w_{t-1}) - \nabla F_m(w_{t-1} - \eta_t \nabla \widetilde{F}_m(w_{t-1})) \right\|_2 \cdot \left\| \eta_t(g_t + \nabla \widetilde{F}_m(w_{t-1})) \right\|_2,$$

$$\geq^{\text{(Ass. 1)}} \left\langle \nabla F_m(w_{t-1}), \eta_t(g_t + \nabla \widetilde{F}_m(w_{t-1})) \right\rangle - \eta_t^2 H \left\| \nabla \widetilde{F}_m(w_{t-1}) \right\|_2 \cdot \left\| g_t + \nabla \widetilde{F}_m(w_{t-1}) \right\|_2,$$

$$\geq \left\langle \nabla F_m(w_{t-1}), \eta_t(g_t + \nabla \widetilde{F}_m(w_{t-1})) \right\rangle - 2\eta_t^2 H,$$

$$= \eta_t \left\| \nabla F_m(w_{t-1}) \right\|_2 \left( 1 + \left\langle \nabla \widetilde{F}_m(w_{t-1}), g_t \right\rangle \right) - 2\eta_t^2 H,$$

$$\geq -2\eta_t^2 H.$$

Choosing $\eta_t \leq \sqrt{\frac{\epsilon}{2H}}$ ensures that,

$$F_m(w_t) >^{\text{(lemma 6)}} 2\epsilon - \epsilon = \epsilon,$$

and thus agent $m$ doesn't defect. $\qquad\square$

### C.4 Proof of Lemma 3

We first introduce the following lemma.

**Lemma 7** (Progress in case 1 and 2). *Under the same conditions as in Theorem 1. Suppose the algorithm is in case 1 or 2 at any time step $t$, and no agent has defected up to time step $t$. If $\eta \leq 1/(MH)$, then $F(w_t) < F(w_{t-1}) - \frac{\eta_t}{2} \min(\|\nabla F(w_{t-1})\|_2^2, 1)$ in case 2 and $F(w_t) < F(w_{t-1}) - \frac{\eta_t}{2M} \min(\|\nabla F_{ND}(w_{t-1})\|_2^2, 1)$ in case 1.*

By Lemma 7, we show that the algorithm makes non-zero progress on the average loss every time it is in case 1 and case 2. This implies that Algorithm 1 will converge. Now we only need to prove that the average loss will converge to 0. In this case, if the algorithm never terminates, then there must be a time step $t$ such that $F_m(w_t) < 2\epsilon$ for all $m \in [M]$. We get into case 3 and then terminate.

We can prove this by contradiction. Suppose that the average loss will converge to $F(w^*) + v = v$ for some $v > 0$. That is to say, for any subsequence, $F(w_t)$ converges to $v$. Then we list all possible subsequences as follows.

1. For a subsequence where the algorithm is in case 2, we will make at least progress scaled with $\|\nabla F(w_{t-1})\|_2^2$. It implies that $\|\nabla F(w_{t-1})\|_2$ would converge to zero for $t$ in this subsequence. Thus applying convexity of $F$ we get that
$$F(w_{t-1}) \leq F(w^*) + \nabla F(w_{t-1})^\top (w_{t-1} - w^*) \to F(w^*) = 0.$$

This is a contradiction.

2. For any subset $S$ of $[M]$, for the subsequence where the algorithm is in case 1 and the predicted non-defecting set is $ND = S$, we will make at least progress scaled with $\|\Pi_P (\nabla F_{ND}(w_{t-1}))\|_2^2$. It implies that $\|\Pi_P (\nabla F_{ND}(w_{t-1}))\|_2^2$ would converge to zero for $t$ in this subsequence. There are two possible cases:

   - Let $\|\nabla F_{ND}(w_{t-1})\|_2^2$ also converge to zero for this sub-sequence. In this case again applying convexity of $F_{ND}$ we get that,

$$F_{ND}(w_{t-1}) \le F_{ND}(w^\star) + \nabla F_{ND}(w_{t-1})^\top (w_{t-1} - w^\star) \to F(w^\star) = 0$$

   converges to zero. This is again a contradiction.
   - Let $\|\nabla F_{ND}(w_{t-1})\|_2^2$ not converge to zero for this subsequence. Then this would violate assumption 4, as everywhere outside the set $\mathcal{W}^\star$, $\nabla F_{ND}(w_{t-1})$ must have a non-zero component in $P$.

Therefore, the algorithm can not converge to a point with a function value $F(w^*) + v = v$. We are done with the proof.

### C.5 PROOF OF LEMMA 4

*Proof.* Let's say that Algorithm 1 terminates in Case 3 at time $t$ then we have for all $m \in D = [M]$,

$$
\begin{aligned}
3\epsilon &\ge^{\text{(Lemma 6)}} F_m(w_{t-1} - \eta_t \nabla \widetilde{F}_m(w_{t-1})), \\
&\ge^{\text{(Convexity)}} F_m(w_{t-1}) + \left\langle \nabla F_m(w_{t-1}), -\eta_t \nabla \widetilde{F}_m(w_{t-1}) \right\rangle, \\
&= F_m(w_{t-1}) + \left\langle \nabla F_m(w_{t-1}), -\eta_t \frac{\nabla F_m(w_{t-1})}{\|\nabla F_m(w_{t-1})\|_2} \right\rangle, \\
&= F_m(w_{t-1}) - \eta_t \|\nabla F_m(w_{t-1})\|_2, \\
&\ge^{(Ass.2)} F_m(w_{t-1}) - \eta_t L.
\end{aligned}
$$

Assuming $\eta_t \le \frac{\epsilon}{L}$ we get that for all $m \in [M]$,

$$
\begin{aligned}
F_m(w_{t-1}) &\le 3\epsilon + \epsilon, \\
&= 4\epsilon,
\end{aligned}
$$

which proves the claim. $\qquad\square$

### C.6 PROOF OF LEMMA 5

*Proof.* Consider some point $w \in \mathcal{W} \setminus \mathcal{W}^\star$ and let the non-zero gradients on the agents be linearly independent at that point. If possible, let the above property be violated, then we have at least one $A \subseteq [M]$ such that $\nabla F_A(w) \in Span\{\nabla F_m(w) : m \in B\}$ and $\nabla F_A \ne 0$. In particular there are coefficients $\{\gamma_n \in \mathbb{R}\}_{n \in B}$ (not all zero) such that,

$$
\begin{aligned}
&\sum_{m \in A} \nabla F_m(w) = \sum_{n \in B} \gamma_n \nabla F_n(w), \\
\Leftrightarrow &\sum_{m \in A} \nabla F_m(w) + \sum_{n \in B} (-\gamma_n) \nabla F_n(w) = 0,
\end{aligned}
$$

This implies that the gradients are linearly dependent (note that not all gradients can be zero as then $w$ would be in $\mathcal{W}^\star$), which is a contradiction. $\qquad\square$

### C.7 PROOF OF LEMMA 6

*Proof.* Let's say we are at time step $t$. Assume $m \in D$ and note that the smoothness of function $F_m$ implies that,

$$F_m(w_{t-1} - \eta_t \nabla \widetilde{F}_m(w_{t-1})) \le F_m(w_{t-1}) + \left\langle \nabla F_m(w_{t-1}), -\eta_t \nabla \widetilde{F}_m(w_{t-1}) \right\rangle + \frac{H}{2} \left\| \eta_t \nabla \widetilde{F}_m(w_{t-1}) \right\|_2^2,$$

$$= F_m(w_{t-1}) - \eta_t \|\nabla F_m(w_{t-1})\|_2 + \frac{H\eta_t^2}{2},$$

$$\leq^{(m \in D)} 2\epsilon + \frac{H\eta_t^2}{2},$$

$$\leq 3\epsilon,$$

where we used that $\eta_t \leq \sqrt{\frac{2}{H}}$. Now assume $m \in ND$ and note using convexity of $F_m$ that,

$$F_m(w_{t-1} - \eta_t \nabla \widetilde{F}_m(w_{t-1})) \geq F_m(w_{t-1}) + \left\langle \nabla F_m(w_{t-1}), -\eta_t \nabla \widetilde{F}_m(w_{t-1}) \right\rangle,$$

$$= F_m(w_{t-1}) - \eta_t \|\nabla F_m(w_{t-1})\|_2,$$

$$>^{(m \in ND)} 2\epsilon.$$

This proves the lemma. $\qquad\square$

### C.8 Proof of Lemma 7

*Proof.* We first assume we make an update in **case 1**. First, using the smoothness assumption and then using the fact that $g_t$ is orthogonal to the gradients of all the agents in $D$, we get

$$F(w_t) = F(w_{t-1} + \eta g_t),$$

$$\leq^{(\text{ass. 1})} F(w_{t-1}) + \eta \left\langle \frac{\sum_{m \in D} \nabla F_m(w_{t-1}) + \nabla F_{ND}(w_{t-1})}{M}, g_t \right\rangle + \frac{H\eta^2}{2} \|g_t\|_2^2,$$

$$\leq F(w_{t-1}) - \frac{\eta}{M} \left\langle \nabla F_{ND}(w_{t-1}), g_t \right\rangle + \frac{H\eta^2}{2} \|g_t\|_2^2,$$

Now we will consider two cases. In the first case, assume $\|\nabla F_{ND}(w_{t-1})\|_2 < 1$. Then we get that,

$$F(w_t) \leq F(w_{t-1}) - \frac{\eta}{M} \left\langle \nabla F_{ND}(w_{t-1}), \nabla F_{ND}(w_{t-1}) \right\rangle + \frac{H\eta^2}{2} \|\nabla F_{ND}(w_{t-1})\|_2^2,$$

$$\leq F(w_{t-1}) - \frac{\eta}{M} \|\nabla F_{ND}(w_{t-1})\|_2^2 + \frac{H\eta^2}{2} \|\nabla F_{ND}(w_{t-1})\|_2^2,$$

$$\leq F(w_{t-1}) - \frac{\eta_t}{2M} \|\nabla F_{ND}(w_{t-1})\|_2^2, \tag{3}$$

where we assume $\eta_t \leq \frac{1}{MH}$. Since in case 1, $\|\nabla F_{ND}(w_{t-1})\|_2 \neq 0$, we will probably make progress on the average objective. In the second case, assume $\|\nabla F_{ND}(w_{t-1})\|_2 \geq 1$. Then we will get that,

$$F(w_t) \leq F(w_{t-1}) - \frac{\eta_t}{M} \|\nabla F_{ND}(w_{t-1})\|_2 + \frac{H\eta_t^2}{2},$$

$$\leq F(w_{t-1}) - \frac{\eta_t}{M} + \frac{H\eta_t^2}{2},$$

$$\leq F(w_{t-1}) - \frac{\eta_t}{2M},$$

where we assume $\eta_t \leq \frac{1}{MH}$. This finishes the proof.

Next, we assume we make an update in **case 2**. Note that using smoothness,

$$F(w_t) = F(w_{t-1} + \eta_t g_t),$$

$$\leq F(w_{t-1}) + \eta_t \left\langle \nabla F(w_{t-1}), g_t \right\rangle + \frac{H\eta_t^2}{2} \|g_t\|_2^2.$$

Let's first assume $\|\nabla F(w_{t-1})\|_2 \geq 1$. Using the definition of $g_t$ we get,

$$F(w_t) \leq F(w_{t-1}) - \eta_t \left\langle \nabla F(w_{t-1}), \frac{\nabla F(w_{t-1})}{\|\nabla F(w_{t-1})\|_2} \right\rangle + \frac{H\eta_t^2}{2} \|g_t\|_2^2,$$

$$= F(w_{t-1}) - \eta_t \|\nabla F(w_{t-1})\|_2 + \frac{H\eta_t^2}{2},$$

$$\leq F(w_{t-1}) - \eta_t + \frac{H\eta_t^2}{2},$$

$$\leq F(w_{t-1}) - \frac{\eta_t}{2},$$

where we assume $\eta_t \leq \frac{1}{H}$. Now let's consider the case when $\|\nabla F(w_{t-1})\|_2 < 1$.

$$F(w_t) \leq F(w_{t-1}) - \eta_t \langle \nabla F(w_{t-1}), \nabla F(w_{t-1}) \rangle + \frac{H\eta_t^2}{2} \|\nabla F(w_{t-1})\|_2^2,$$

$$= F(w_{t-1}) - \eta_t \|\nabla F(w_{t-1})\|_2^2 + \frac{H\eta_t^2}{2} \|\nabla F(w_{t-1})\|_2^2,$$

$$\leq F(w_{t-1}) - \frac{\eta_t}{2} \|\nabla F(w_{t-1})\|_2^2, \tag{4}$$

where we assume $\eta_t \leq \frac{1}{H}$. Note that $\|\nabla F(w_{t-1})\|_2 \neq 0$ in this case, which means the algorithm makes non-zero progress on the average objective. $\qquad\square$

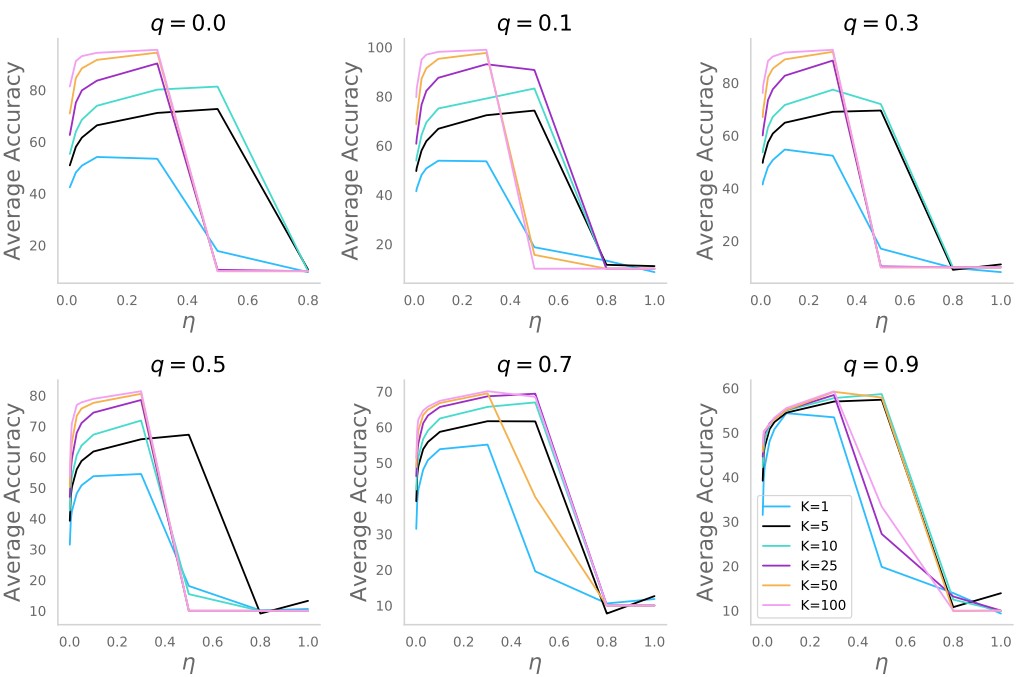

Figure 8: Fine-tuning the step-size $\eta$ for different data heterogeneity $q$ (across different plots) and the number of local update steps $K$ (different curves in each plot). The required precision $\epsilon = 0$ during the fine-tuning phase.

## D MORE DETAILS ON THE EXPERIMENTS

**Generating data with heterogeneity** $q$. Denote the dataset $\mathcal{D} = \{\mathcal{D}_1, \cdots, \mathcal{D}_n\}$ where $n$ is the number of devices. To create a dataset with heterogeneity $q \in [0, 1]$ for every device, we first pre-process the dataset of every device such that $|\mathcal{D}_1| = \cdots = |\mathcal{D}_n|$. Then for every device $i$, we let that device keep $(1 - q) \cdot \mathcal{D}_i$ samples from their own dataset and generate a union dataset $\widehat{\mathcal{D}}$ with the remaining samples from all devices, i.e. $\widehat{\mathcal{D}} = q \cdot \mathcal{D}_1 \cup \cdots \cup q \cdot \mathcal{D}_n$. We use $q \cdot \mathcal{D}_i$ to denote a random split of $q$ portion from the dataset $\mathcal{D}_i$. Finally, the data with heterogeneity $q$ for every device $i$ is generated by

$$\widehat{\mathcal{D}}_i = (1 - q) \cdot \mathcal{D}_i \cup \frac{1}{n} \cdot \widehat{\mathcal{D}}$$

**Additional experimental results.** We present our fine-tuning process for finding the step size for different settings in Figure 8. In Figures 9 —11, we also present additional findings with more variations in data heterogeneity $q$ and the number of local update steps $K$ besides the results we presented in Figure 3 of the main paper.

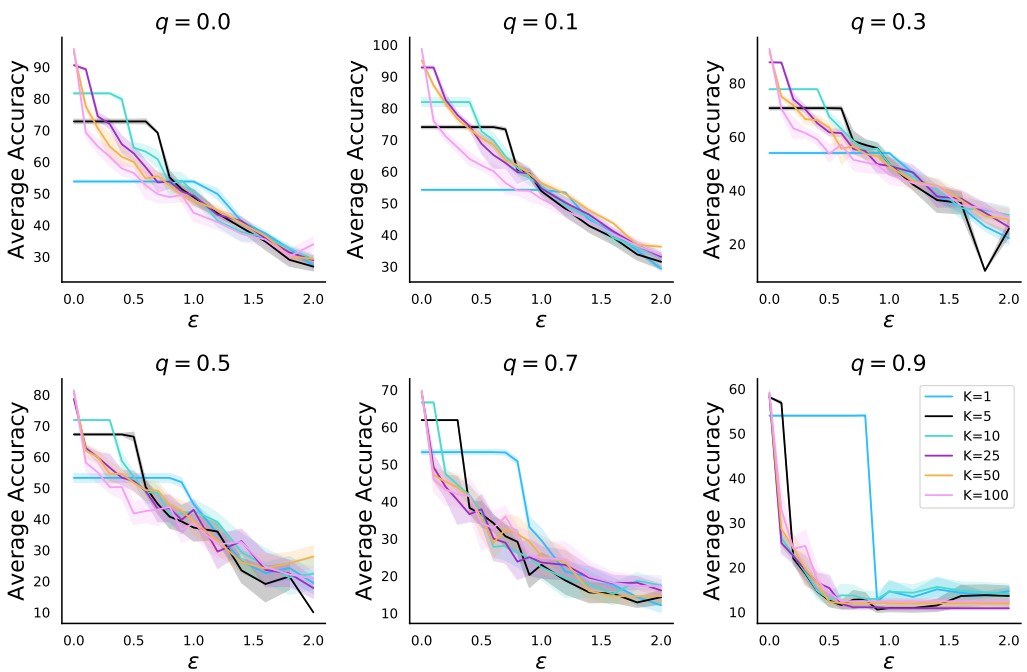

Figure 9: Additional findings on the effect of defection on average accuracy

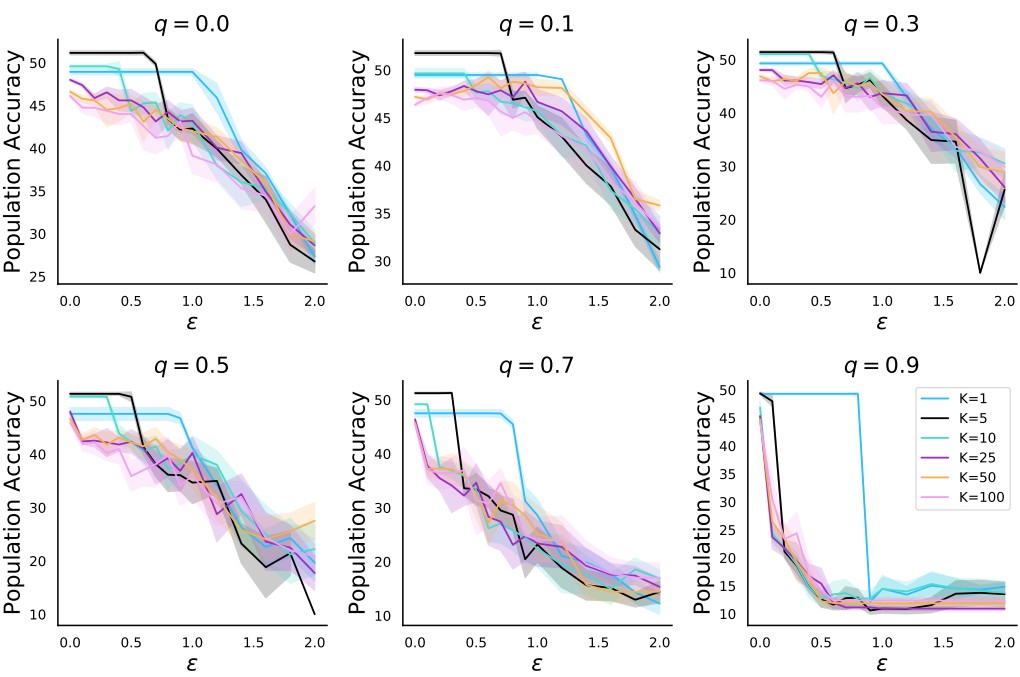

Figure 10: Additional findings on the effect of defection on population accuracy

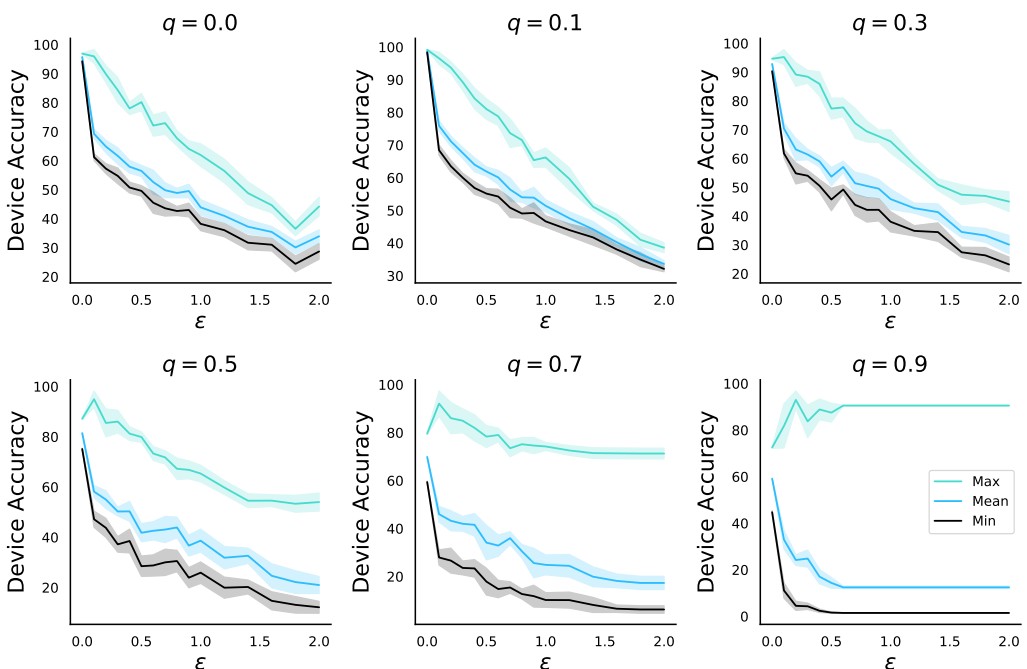

Figure 11: Additional findings on the effect of defection on the min, mean, and max device accuracies

## E  OPEN QUESTIONS AND FUTURE WORK

In this paper, we studied the impact of agents' defections in federated learning and proposed an algorithm to avoid defections. As aforementioned in the section of Discussion, one open problem is how to extend our Algorithm 1 when only stochastic oracles are accessible. There are some other interesting future directions in the following.

**Convergence rate.**  We only provide asymptotic convergence guarantee for Algorithm 1. The non-asymptotic guarantee is an interesting open question. The main difficulty lies in that there are several phases and we analyze each phase separately. The standard technique for analyzing first-order methods cannot be applied in this case. Furthermore, even in the asymptotic setting, we believe we can improve our precision guarantee from $4\epsilon$ to $(1 + \delta)\epsilon$ for arbitrarily small $\delta$.

**Approximately realizable setting.**  In this work, we focus on the realizable setting, where there exists $w^* \in \mathcal{W}$ such that $F_m(w^*) = 0$ for all $m \in [M]$. However, the next natural question to ask is: can we get similar results when $w^*$ is only approximately optimal for all agents?

**Non-convex optimization.**  We focus on the convex optimization setting. However, a natural follow-up question is how to avoid defections in non-convex federated optimization. While we perform experiments in the non-convex setting, our theory doesn't capture this setting.

**Other rational behaviors.**  We restrict to the setting where the server cannot save the intermediate models and wants the final model to be as good as possible. It is also interesting to consider settings where the server can save multiple intermediate models and use the best one when a new agent from the population arrives.

