# OpenReview forum: "On the Effect of Defection in Federated Learning and How to Prevent It"
_ICLR.cc/2024/Conference — Submitted to ICLR 2024_

### Official Review · Reviewer_bhy3 · 2023-10-29

**Soundness:** 1 poor
**Presentation:** 2 fair
**Contribution:** 1 poor
**Rating:** 3
**Confidence:** 4

**Summary:**

The paper studies federated learning problems where the agents can strategically choose the early quit the FL process. The authors present examples when such early quitting can happen and how these cases can affect the losses of all agents. The authors later present an algorithm: Adaptive Defection-aware Aggregation for Gradient Descent (ADA-GD) and claim that this algorithm can disincentivize early quitting under some assumptions on the loss functions, parameter space, and the agents' heterogeneity.

**Strengths:**

The authors are able to make several observations on why the defective early quitting can happen and identified cases when such quitting may or may not negatively influence the agents that have not quitted. Overall, the problem of preventing early quitting is an interesting branch in strategic FL problems.

**Weaknesses:**

1. Unrealistic problem setting on the agents' incentives. The authors assume that the agents quit the FL process as long as they are \epsilon close to optimal solution. In ADA-GD, the agents are simply ignored when they are sufficiently close to the \epsilon-optimal set and are kept waiting in the system. This is highly unrealistic, if the faster agents know the server will implement ADA-GD, then the very first time they are ignored, why don't they early quit and run another round of local gradient update? The assumption that the agents are only first-order strategic is not a good enough assumption, there should be more careful design on the backward induction steps, and I think we need a much more sophisticated algorithm than ADA-GD to make sure strategic agents are happy (incentive compatible and individually rational) to participate in the algorithm given they have realistic outside options (like training the final gradient step themselves).
2. Unrealistic assumptions on the server's information availability. In ADA-GD, the server can observe the agents' losses instead of just their gradients is not realistic.
3. In sufficient discussion on the field of FL, strategic learning and insufficient comparisons with related works, I suggest the authors provide a table that list out the (1) type of strategic manipulations of related works in FL, as well as the (2) assumptions those paper make, and (3) convergence as well as robustness guarantees. Moreover, I'm very suspicious about the claim on page 4 "if there is no shared minima for all agents, applying FL is not reasonable". First of all, this is not known prior to participating in the FL process for all agents. Secondly, I suggest the authors discuss related works in personalized FL and further explain this claim here.
4. Very restrictive settings like "strong convexity, smoothness, realizability, and minimal heterogeneity" makes the algorithm unable to run on deep learning tasks like vision and NLP.
5. Unclear why the server wants to find a solution in W^*, as long as the agents are all \epsilon happy, all participants should be fine with the outcome.

**Questions:**

Please refer to the weaknesses part and explain why the unrealistic claims are in fact realistic.

In addition,
1. What is the model used for 2 class Cifar-10 classification? Why does this satisfy all the assumptions in the paper?
2. Which of the examples satisfy Assumption 4?

---

> ### Author Response · Authors · 2023-11-19
> **Response 1/2**
>
> We thank the reviewer for their comments and address their concerns below. We hope the reviewer will re-consider their score.
>
> > Unrealistic problem setting on the agents' incentives. The authors assume that the agents quit the FL process as long as they are \epsilon close to optimal solution. In ADA-GD, the agents are simply ignored when they are sufficiently close to the \epsilon-optimal set and are kept waiting in the system. This is highly unrealistic, if the faster agents know the server will implement ADA-GD, then the very first time they are ignored, why don't they early quit and run another round of local gradient update? The assumption that the agents are only first-order strategic is not a good enough assumption, there should be more careful design on the backward induction steps, and I think we need a much more sophisticated algorithm than ADA-GD to make sure strategic agents are happy (incentive compatible and individually rational) to participate in the algorithm given they have realistic outside options (like training the final gradient step themselves).
>
>
>
> “the very first time they are ignored, why don't they early quit and run another round of local gradient update” is an interesting question. In this work, we restrict the action space of agents to “deciding if they want to quit”. The reviewer suggests broadening this action space to include “deciding both on quitting and running additional rounds”. This will make avoiding defections much more challenging.
>
> While we acknowledge this as an intriguing avenue for future exploration, our work is an initial step in comprehending agent defections in Federated Learning. One primary contribution of our study demonstrates that even within the simple action space of 'deciding whether to quit,' the widely used FedAvg method fails. Conversely, our algorithm proves effective. The next interesting question, as hinted by the reviewer, is: "How can we prevent defections when the workers' action space becomes more complex?"
>
> Having said that, we encourage the reviewer to read our response to reviewer sato. It is possible to incorporate strategizing during local update steps, i.e., agents can leave the collaboration within the communication round. If the algorithm wants to be conservative against further strategizing, the rule in line 4 of the algorithm can be made more stringent. The natural trade-off here is that the more stringent the rule to detect the defecting agents, the worse the guarantee of the final model will be.
>
>
> > Unrealistic assumptions on the server's information availability. In ADA-GD, the server can observe the agents' losses instead of just their gradients is not realistic.
>
> We would like first to address that this is the standard definition of a first-order oracle—zeroth and first-order information. In most deep learning implementations, obtaining the loss of the model every few rounds is standard, and most optimization procedures have their stopping criterion based on the current loss. This is in no way contrary to the goal of federated learning, which only prohibits sharing raw data. If we were to consider the setting where the algorithms have gradient information only, there would be no way for the server to track how close a client is to meeting its goal: a client with a flat loss function might be very far from its optima, while a client with a very sharp function might be very close to it. Can the reviewer highlight why they think access to loss is unrealistic?
>
> We want to highlight the negative result of this work: FedAvg's inability to prevent harmful defections and the necessity of assumption 4 (minimal heterogeneity) for averting harmful defections in small step-size ICFO algorithms (as illustrated in Figure 6). While our algorithm relies on loss information, it can be a building block for developing algorithms in a broader context.
>
> > In sufficient discussion on the field of FL, strategic learning and insufficient comparisons with related works, I suggest the authors provide a table that list out the (1) type of strategic manipulations of related works in FL, as well as the (2) assumptions those paper make, and (3) convergence as well as robustness guarantees.
>
> Thanks for the suggestion! We will add such a table. Please also see the response to reviewer y9Cn regarding related work and our three-page literature review in the appendix.

---

> ### Author Response · Authors · 2023-11-19
> **Response 2/2**
>
> > Moreover, I'm very suspicious about the claim on page 4 "if there is no shared minima for all agents, applying FL is not reasonable". First of all, this is not known prior to participating in the FL process for all agents. Secondly, I suggest the authors discuss related works in personalized FL and further explain this claim here.
>
> We did not intend to say that applying FL is not reasonable if there is no shared optima for all agents. Indeed, personalized approaches can still benefit from collaboration. However, the vanilla FL objective we discuss and define in the paper (c.f., objective (2)) uses the same consensus model for all the agents. Without a shared optimum, collaboration (instead of training a local model) is not guaranteed to benefit the agents due to data heterogeneity. While this information is not known a priori, this paper aims to start with a simple, clean setting where joining collaboration can guarantee a good consensus model. We could have considered relaxations to this assumption, such as bounded data heterogeneity assumptions, but we strongly believe that it would only obfuscate the paper's main message. We will clarify this discussion in our final version.
>
> > Very restrictive settings like "strong convexity, smoothness, realizability, and minimal heterogeneity" makes the algorithm unable to run on deep learning tasks like vision and NLP.
>
> These assumptions are common in theoretical analysis, and for deep learning, we run experiments to justify the effectiveness of our algorithm. Specifically, our empirical validation of MNIST and CIFAR-10 with fully connected neural networks substantiates our theoretical analysis. Note that we do not assume strong convexity. We would like to highlight again that we do not know of any other theoretical work that explicitly and provably prevents defection. We start with tractable assumptions, and as we discuss in the paper, the problem is hard for existing algorithms, thus motivating additional assumptions like minimal heterogeneity. We will explore more relaxed settings in future work.
>
> > Unclear why the server wants to find a solution in W^*, as long as the agents are all \epsilon happy, all participants should be fine with the outcome.
>
> To clarify (see discussion below assumption 3 on page 5), our goal is to make all the agents $\epsilon$ happy instead of finding a solution in $W^*$. We introduce $W^*$ because a model that is $\epsilon$ good for every agent is close to $W^*$. The server wants to find a good model for all agents because our original learning goal (explained through the hospital example) is (1). A model that works well for all agents participating in the optimization will likely have a low population loss.
>
> > What is the model used for 2 class Cifar-10 classification? Why does this satisfy all the assumptions in the paper?
>
> We run empirical validation of MNIST and CIFAR-10 with a (two-layer) fully connected neural network, they are mainly for empirically validating that our algorithm can indeed run on deep learning tasks like vision and NLP, as questioned by the reviewer as well.
>
> > Which of the examples satisfy Assumption 4?
>
> The examples we gave in the papers serve as counterexamples, illustrating the outcomes when our specified assumptions are not met. For an example that satisfies assumption 4, please refer to [this anonymized figure](https://anonymous.4open.science/r/IncentivesFL-E225/intersection.png).

---

### Official Review · Reviewer_VX8w · 2023-10-30

**Soundness:** 3 good
**Presentation:** 3 good
**Contribution:** 2 fair
**Rating:** 6
**Confidence:** 3

**Summary:**

This paper studied the problem of defections in federated learning, where agents may choose to defect permanently, i.e. withdrawing from the collaboration, if they are content with their instantaneous model in a round. The paper first analyzed the potential negative impact of such defection on the final model's robustness and ability to generalize. It distinguished between benign and harmful defections and explore the influence of (i) initial conditions (ii) learning rates (iii) aggregation methods on the occurrence of harmful defections. The paper then proposed a new algorithm which prevents defection and analyzed its properties theoretically and empirically (in comparison to FedAvg).

**Strengths:**

- Overall this paper is written with a good clarity, and the illustrative examples are helpful for understanding the mechanisms / possible impact of defection
- The paper studies an important and practical game theoretical consideration in federated learning, where the agents are not always incentivized to stay. Through detailed examples with two agents, the paper provided nice examples where the defections can be benign or harmful depending on the different initializations, learning rates or aggregation algorithms.
- The proposed algorithm offers an intuitive and natural solution, where the server simply tunes the update direction to avoid the defection of any agent which is near the defection threshold. Theoretically under assumptions 1-4, this algorithm is guaranteed to prevent defection.
- Numerical simulations in comparison to FedAvg are provided to support the theoretical result.

**Weaknesses:**

- Under the current algorithm design, it seems that Assumption 4 (Minimal Heterogeneity) is rather crucial. However, such an assumption which essentially assumes no "similar" agents seem to be hard to achieve in practice, in particular agents tend to join federated learning if they share similar goals and have similar loss functions to minimize. It would be good to see more discussion how deviation from the perfect heterogeneity can impact the proposed algorithm's outcome.
- The simulations presented in the paper were relatively week and were conducted in simple settings with only two agents. The empirical evaluations need to be strengthened with larger number of agents / more realistic datasets.

**Questions:**

- Will algorithm 1 incentivize the agents that are on the verge of defecting to report non-truthful losses to the server?
- How does the violation of minimal heterogeneity affect the guarantee of algorithm 1?

---

> ### Author Response · Authors · 2023-11-18
>
> We thank the reviewer for their comments and address their concerns about assumption 4 and experiments below.
>
> > Under the current algorithm design, it seems that Assumption 4 (Minimal Heterogeneity) is rather crucial. However, such an assumption which essentially assumes no "similar" agents seem to be hard to achieve in practice, in particular agents tend to join federated learning if they share similar goals and have similar loss functions to minimize. It would be good to see more discussion how deviation from the perfect heterogeneity can impact the proposed algorithm's outcome
>
> > How does the violation of minimal heterogeneity affect the guarantee of algorithm 1?
>
> In Figure 6 on page 7, we presented an example that doesn't meet assumption 4 and concluded that no algorithm employing small step sizes can prevent harmful defections. Essentially, it is crucial to assume the presence of a continuous descent path from the initialization to the target region that doesn't cross the $\epsilon$-sublevel set of any agent. Without this assumption, no gradient descent-based algorithm can prevent defections. Our assumption ensures the existence of such a path. We appreciate the reviewer for highlighting this aspect and plan to expand on this in our paper.
>
> > The simulations presented in the paper were relatively week and were conducted in simple settings with only two agents. The empirical evaluations need to be strengthened with larger number of agents / more realistic datasets.
>
> Our proposed method works for more than two agents and $K>1$.  We have conducted one additional experiment, please refer to [this anonymized figure](https://anonymous.4open.science/r/iclr_anony-3E5D/additioanl_results.png) with $10$ agents and $K=5$. We are running more extensive experiments and will include them in the camera-ready version of our paper.
>
>
> > Will algorithm 1 incentivize the agents that are on the verge of defecting to report non-truthful losses to the server?
>
> This is an excellent question. This work assumes that all agents will report truthful first-order information. If the agents are allowed to report non-truthful losses, then every agent can tell the server their loss is maximal, and our algorithm 1 will degenerate to FedAvg; hence, we can’t avoid defections. The problem will become much more complicated if the agents can tell non-truthful losses. In future work, we would like to explore this direction to ensure that truthful reporting is indeed incentive-compatible.
>
> We hope the reviewer will reconsider their score.

---

### Official Review · Reviewer_sato · 2023-10-30

**Soundness:** 3 good
**Presentation:** 3 good
**Contribution:** 2 fair
**Rating:** 3
**Confidence:** 4

**Summary:**

The paper discusses when agents in a federated system will defect, and proposes a novel aggregation strategy to prevent defection.
Defection occurs when any particular agent achieves an $\epsilon$-optimal answer and quits from the federation system.
This paper demonstrates how the occurrence of defection is related to choices of initialization and learning rates.
Moreover, the authors propose an example to show that defection is inevitable for the strategy of uniform aggregation.
Instead of uniform aggregation, the paper proposes a novel strategy that predicts whether an agent will defect in the next round and projects the aggregated direction to the orthogonal space of its gradient.
It is claimed that no agent will defect under the novel aggregation strategy.

**Strengths:**

1. The paper considers the quit of agents in federated systems, which is missing in previous works of federated learning. Agents stop contributing their data when their local models achieve the $\epsilon$-optimal set of their local problems. The quitting mechanism naturally matches most real-life situations.
2. The paper discusses the effects of defection in detail with an easy but clear example. Moreover, a counterexample is proposed to show failures of the uniform aggregation in avoiding defection.

**Weaknesses:**

1. The quitting mechanism introduced in 'Rational Agents' is weird. Specifically, an agent only considers quitting after communication rounds while it does not consider quitting during local updates.
2. The empirical experiment is not enough to justify the proposed method. Firstly, $K=1$ is not enough for the application of federated learning. Most applications of federated learning are carried out with a larger number of local updates, $K>1$.
3. The performance of ADA-GD is not comparable with local SGD using uniform aggregation. Figure 7(a) reveals that local SGD achieves higher accuracy and lower error before the defection happens. In this way, I believe that uniform aggregation with a controller detecting the defection for an early stop will beat ADA-GD.

**Questions:**

See Weaknesses and there are some typos as follows:
1. Why is the updating direction of $w_2$ in Figure 5(a) not parallel with those of $w_1,w_3,w_4$?
2. What do you mean with $(2\epsilon,\epsilon)$ in explaining Observation 3?
3. The definition of $\nabla F(w_{t-1})$ is missing in Algorithm 1.

---

> ### Author Response · Authors · 2023-11-18
>
> We thank the reviewer for their comments. We address the reviewer’s main concerns about $K>1$ local steps in our theory and experiments below. We hope the reviewer will reconsider their score.
>
> > The quitting mechanism introduced in 'Rational Agents' is weird. Specifically, an agent only considers quitting after communication rounds while it does not consider quitting during local updates.
>
> First, note that we demonstrate that FedAvg fails to prevent harmful defections, **even in the simple scenario where agents only quit after communication and not during local update steps**. Thus, considering the more complicated quitting mechanism proposed by the reviewer will only make preventing defections more challenging for existing algorithms. On the other hand, our method can easily be extended to the setting where the agents can potentially defect during the local steps.
>
> To do so, we must change the rule predicting the defecting agents in line 4. We require a more conservative lower bound on the function value on the agent after $K>1$ local updates. We can get such a lower bound by noting that the Hessian is positive semi-definite for convex functions. This implies that the per-step improvement in function value decreases as we approach the optima (on a descent path). Now, as long as this conservative lower bound of the function value after $K$ steps is larger than the defection threshold for that agent, this agent will provably not defect during local updates. In our paper, we chose to discuss only $K=1$ as the main idea of our algorithm can succinctly be described in that simpler setting. We will add a remark discussing the quitting mechanism proposed by the reviewer.
>
> > The empirical experiment is not enough to justify the proposed method. Firstly, $K=1$ is not enough for the application of federated learning. Most applications of federated learning are carried out with a larger number of local updates, $K>1$.
>
> Our proposed method works for more than two agents and $K>1$.  We have conducted one additional experiment, please refer to [this anonymized figure](https://anonymous.4open.science/r/iclr_anony-3E5D/additioanl_results.png).
>
> > The performance of ADA-GD is not comparable with local SGD using uniform aggregation. Figure 7(a) reveals that local SGD achieves higher accuracy and lower error before the defection happens. In this way, I believe that uniform aggregation with a controller detecting the defection for an early stop will beat ADA-GD.
>
> We respectfully disagree. As we have already mentioned in the caption of Figure 7, “it is unfair to compare the highest accuracy of Local SGD with our method as Local SGD is not defection aware and could not simply stop at the highest point.” Specifically, it will keep running without knowing any agent could defect in the next round, thus leading to poor accuracy. To illustrate this, consider a scenario with frequent defections. In such cases, determining the optimal stopping point becomes challenging. Regardless, in our new experiment (mentioned above), even the highest point of local SGD is worse than ADA-GD.
>
> > See Weaknesses and there are some typos as follows…
>
> Thanks, we will address these typos in our revision.

---

### Official Review · Reviewer_y9Cn · 2023-11-01

**Soundness:** 2 fair
**Presentation:** 2 fair
**Contribution:** 2 fair
**Rating:** 3
**Confidence:** 4

**Summary:**

This paper studies possible defection in federated learning where clients can choose to opt out, which can negatively affect the final model performance. This paper proposes a new optimization algorithm that aggregates the clients differently to avoid defections and achieve convergence.

**Strengths:**

- The investigated topic is interesting and relevant for current FL algorithms.
- Theoretical analysis is provided for smooth and convex problems.
- Provide simple experiments to show that the proposed algorithms outperforms the previous algorithms.

**Weaknesses:**

- A concern I have is that ADA-GD assumes that clients are required to be rational in the way that ADA-GD defines them to be. In other words, the fixed precision parameter $\epsilon$ is the universal learning goal which may not be practical in practice where clients can have heterogeneous goals. The proposed definition of the rational agents is rather ambiguous, and also how to set the precision parameters is tricky. What does it really mean for a potentially defecting worker to be content in practice?

- Another concern I have is the way the ADA-GD excludes updates from the defecting workers. Wouldn't this lead to a biased model towards non-defecting clients?

- Also, I wonder that if the workers have the freedom opt in or opt out, wouldn't the server not be able to force the clients to participate?

- Lastly, is there a reason that the work has not compared with other incentivized FL work such as [1-3] below?
  [1] Yae Jee Cho, Divyansh Jhunjhunwala, Tian Li, Virginia Smith, and Gauri Joshi. To federate or not to federate: Incentivizing client participation in federated learning. arXiv preprint arXiv:2205.14840, 2022.
  [2] Avrim Blum, Nika Haghtalab, Richard Lanas Phillips, and Han Shao. One for one, or all for all: Equilibria and optimality of collaboration in federated learning. In International Conference on Machine Learning, pp. 1005–1014. PMLR, 2021.
  [3] Rachael Hwee Ling Sim, Yehong Zhang, Mun Choon Chan, and Bryan Kian Hsiang Low. Collaborative machine learning with incentive aware model rewards. In International Conference on Machine Learning, pp. 8927–8936. PMLR, 2020.

I noticed that the authors have referenced the literature but I do not agree with the authors' note that these work are a bit orthogonal to your work. Why is this the case?

**Questions:**

See Weaknesses Above.

---

> ### Author Response · Authors · 2023-11-18
> **Response 1/2**
>
> We thank the reviewer for their review. We address the reviewer's concerns below and hope they will reconsider their score.
>
> > A concern I have is that ADA-GD assumes that clients are required to be rational in the way that ADA-GD defines them to be. In other words, the fixed precision parameter $\epsilon$ is the universal learning goal which may not be practical in practice where clients can have heterogeneous goals. The proposed definition of the rational agents is rather ambiguous, and also how to set the precision parameters is tricky. What does it really mean for a potentially defecting worker to be content in practice?
>
> Please note that it is not mandatory for all clients to have the same precision parameter, $\epsilon$, nor is it necessary for the algorithm to set specific precision parameters. Each client determines their objectives, such as achieving a 90% model accuracy, **a decision that rests with the client rather than the server**. For the sake of clarity in our presentation, we use a uniform $\epsilon$ precision parameter.
>
> If clients have varying goals—for example, some targeting 90% accuracy while others aim for 95%—our algorithm can accommodate these differences. It does so by adjusting $\epsilon$ to a personalized precision parameter $\epsilon_m$ in line 4 of Algorithm 1. This modification allows the algorithm to predict a client's potential defection based on their unique learning objective. Consequently, the final output of our algorithm aims to satisfy each client’s specific goal. Given that clients choose their own learning goals, they tend to be satisfied with the model provided to them. We will include this simple extension as an additional remark under the theorem.
>
>
> > Another concern I have is the way the ADA-GD excludes updates from the defecting workers. Wouldn't this lead to a biased model towards non-defecting clients?
>
> Our ADA-GD algorithm ensures that the resulting model is not biased toward clients who do not defect. It only produces a model (as described in case 3) when it predicts every client is likely to defect. This means that each client already possesses a nearly optimal model for them. We will make this point clear in the theorem's statement. This approach is feasible because ADA-GD dynamically assesses which clients are on the verge of defection at each time step. As a result, if ADA-GD predicts that a client is close to defection in a given round, it will not include that client's update. However, if the same client later shows a significant loss and is deemed less likely to defect, their updates will be included again. This method creates a back-and-forth movement near the edge of the client's $\epsilon$-sublevel set, ensuring that the model remains effective for that client. For a visual representation, please refer to [this anonymized figure](https://anonymous.4open.science/r/IncentivesFL-E225/intersection.png).
>
>
>
> > Also, I wonder that if the workers have the freedom opt in or opt out, wouldn't the server not be able to force the clients to participate?
>
> Compelling one party to act a certain way is difficult in real-world collaborations due to emerging data protection and privacy laws. For instance, in our running example with hospitals, several laws such as [HIPPA](https://www.hhs.gov/hipaa/index.html) and [CMIA](https://consumercal.org/about-cfc/cfc-education-foundation/cfceducation-foundationyour-medical-privacy-rights/confidentiality-of-medical-information-act/) prohibit a hospital from using a patient’s data without prior permission, and the patient can choose to revoke this permission. This would make it hard for any agency to force hospitals to participate. It is true that in some scenarios, the server can enforce contractual obligations. However, the spirit of federated learning is to foster organic collaborations that continually benefit all the participants as opposed to such contractual collaborations.

---

> > ### Author Response · Authors · 2023-11-18
> > **Response 2/2**
> >
> > > Lastly, is there a reason that the work has not compared with other incentivized FL work such as [1-3] below?
> >
> > We note that there are several differences between the settings of these three papers compared to ours. Most importantly none of these papers theoretically consider multi-round interactions between a server and agents where the agents can permanently defect. We discuss these differences more carefully below and will include this discussion in our revision.
> >
> > > [1] Yae Jee Cho, Divyansh Jhunjhunwala, Tian Li, Virginia Smith, and Gauri Joshi. To federate or not to federate: Incentivizing client participation in federated learning. arXiv preprint arXiv:2205.14840, 2022.
> >
> > This paper provides a different objective for federated learning which is claimed to improve the global model’s appeal for all the devices. To do this, the paper explicitly designs an objective that forces the global model to be better than the individual model of an agent. While the proposed method does empirically improve the global model’s appeal, the paper does not have any theoretical guarantees to ensure that agent defection does not happen. To do so, any algorithm must manipulate the model’s performance on different agents during training (like ADA-GD), and not just its final properties (like [1] does). This is why we believe this work is orthogonal and complementary to ours. For instance, we can take the objective proposed in this paper and combine it with the non-uniform aggregation from our paper.
> >
> > > [2] Avrim Blum, Nika Haghtalab, Richard Lanas Phillips, and Han Shao. One for one, or all for all: Equilibria and optimality of collaboration in federated learning. In International Conference on Machine Learning, pp. 1005–1014. PMLR, 2021.
> >
> > We have very carefully studied [2] as it was one of the motivations for our paper. However, the authors for [2] propose a totally different incentive problem compared to our paper. First, they study single-round interaction between the server and the clients. More specifically, before the training, the server computes the amount of the data needed from each client and asks them to contribute the corresponding amount of data. Second, they assume that the loss for each client is a function of each client’s contribution and is known to the server. In our work, we don’t make such an assumption.
> >
> >
> > > [3] Rachael Hwee Ling Sim, Yehong Zhang, Mun Choon Chan, and Bryan Kian Hsiang Low. Collaborative machine learning with incentive aware model rewards. In International Conference on Machine Learning, pp. 8927–8936. PMLR, 2020.
> >
> > A key distinction between our work and this paper is that they primarily investigate single-round interactions among collaborative ML models, emphasizing the valuation of participating agents' models. Their paper introduces a reward scheme and demonstrates that it satisfies certain incentive conditions (e.g., feasibility, individual rationality, and fairness). In contrast, our focus lies in providing theoretical guarantees to prevent defections and ensuring asymptotic convergence to the effective solution for all participating agents in multi-round interactions.

---

### Author Response · Authors · 2023-11-23
**Request to engage as the discussion ends**

Dear Reviewers,



We want to thank you again for your valuable time and comments.

We believe that several of the pointed-out limitations arose from a misunderstanding of our strategic setting and how it compares to existing work. In our response, we have clarified these issues and will add a more detailed comparison to related but orthogonal works. We want to re-emphasize that the problem of agent defection in multi-round FL protocols has **not** been addressed in existing work. We have done a detailed literature review (which can be found in our appendix). While our model is simple, we believe it is a useful starting step towards understanding the effect of defections. We have also argued why permanent defections are well-motivated in our running example of medical collaboration.

While other strategic settings beyond what we consider in this paper are important (such as non-truthful reporting of losses), our result highlights how existing algorithms, such as FedAvg, fail even in the simple setting we consider. We have also provided clarifications about how our results can be adapted to incorporate the limitations pointed out by the reviewers, such as strategizing in the presence of local updates. Finally, we have provided additional [figures](https://anonymous.4open.science/r/IncentivesFL-E225/intersection.png) and [experiments](https://anonymous.4open.science/r/iclr_anony-3E5D/additioanl_results.png) to respond to the limitations pointed out by the reviewers directly. We hope the reviewers will reconsider our algorithm's effectiveness in avoiding defections and obtaining a good model in light of these new experiments.

We are happy to answer more questions. As the discussion period ends, we hope the reviewers will update their initial scores.



Best,

Authors

---

### Meta-Review · Area_Chair_8Qpz · 2023-12-11

**Metareview:**

This paper appears to address a novel aspect of federated learning (FL): the early quitting of agents. While the topic is timely and interesting to the FL community, the reviewers raised several issues. A recurring concern across the reviews is whether the assumptions are realistic, particularly regarding agent behavior and the server's information availability. The method of excluding updates from defecting workers raises concerns about resulting bias. The empirical evaluation is overall weak. A reviewer also suggested that there should be a more in-depth discussion on how this work is related to other prior work on FL and incentives.

**Justification For Why Not Higher Score:**

The results are not sufficiently strong.

**Justification For Why Not Lower Score:**

N/A

---

### Decision · Program_Chairs · 2024-01-16

Reject